# Interhemispheric competition during sleep

Lorenz A. Fenk[1✉], Juan Luis Riquelme[1,2] & Gilles Laurent[1✉]

Our understanding of the functions and mechanisms of sleep remains incomplete, reflecting their increasingly evident complexity[1–3]. Likewise, studies of interhemispheric coordination during sleep[4–6] are often hard to connect precisely to known sleep circuits and mechanisms. Here, by recording from the claustra of sleeping bearded dragons (*Pogona vitticeps*), we show that, although the onsets and offsets of *Pogona* rapid-eye-movement (REM$_P$) and slow-wave sleep are coordinated bilaterally, these two sleep states differ markedly in their inter-claustral coordination. During slow-wave sleep, the claustra produce sharp-wave ripples independently of one another, showing no coordination. By contrast, during REM$_P$ sleep, the potentials produced by the two claustra are precisely coordinated in amplitude and time. These signals, however, are not synchronous: one side leads the other by about 20 ms, with the leading side switching typically once per REM$_P$ episode or in between successive episodes. The leading claustrum expresses the stronger activity, suggesting bilateral competition. This competition does not occur directly between the two claustra or telencephalic hemispheres. Rather, it occurs in the midbrain and depends on the integrity of a GABAergic (γ-aminobutyric-acid-producing) nucleus of the isthmic complex, which exists in all vertebrates and is known in birds to underlie bottom-up attention and gaze control. These results reveal that a winner-take-all-type competition exists between the two sides of the brain of *Pogona*, which originates in the midbrain and has precise consequences for claustrum activity and coordination during REM$_P$ sleep.

In mammals, cortical electroencephalograms during sleep can be decomposed into rapid-eye-movement (REM) sleep, which is characterized by desynchronized electroencephalographic signals accompanied by rapid eye movements[7–9], and non-REM (NREM) sleep, which is characterized by slow-wave activity (and hence is also called slow-wave (SW) sleep). In rodents and humans, NREM sleep has been implicated in the reactivation and consolidation of certain forms of memories[10–13]. The possible functions of REM, however, remain largely speculative (for example, unlearning[14]), although some evidence[15–17] suggests a possible link with emotional memory. Biphasic sleep also exists in birds[18], reptiles[19–21] and fish[22], suggesting the possibility of common roots (at least as old as the vertebrate lineage (500 million years)). (Biphasic sleep patterns have also been observed in some invertebrates[23]). If so, comparative approaches in systems that represent diverse animal lineages might help us to better understand not only the evolution of sleep, but also its functions and mechanistic underpinnings.

In *Pogona vitticeps*, a diurnal agamid lizard, the two phases of a regular electrophysiological sleep rhythm (as recorded in the dorsal-ventricular ridge or DVR) consist of local field potentials dominated by sharp-wave ripples that occur irregularly every 0.5–2 s for about a minute, followed by faster awake-like activity that co-occurs with rapid eye movements[19], also for about a minute. These two activity modes alternate regularly throughout the night[19]. Because of the similarities between these two electrophysiological sleep phases and mammalian SW and REM sleep, we identify them as SW-like and REM-like (thereafter respectively named SW and REM$_P$). (Note that our nomenclature is descriptive, and does not necessarily imply, for

lack of knowledge at this point, functional or mechanistic identity with mammalian sleep states). The dominant features of SW activity in the DVR—the sharp-wave ripples—are produced in the claustrum, and are detectable in the adjacent DVR as a propagating wave[21]. Here we set out to examine the nature of REM$_P$ signals in *Pogona*, using electrophysiological recordings in vivo. Recording from the left and right claustra simultaneously, we observed differences in the nature of interhemispheric coordination between SW and REM$_P$ sleep. These differences, in turn, revealed an ongoing competition between the two hemispheres during REM$_P$ sleep—but not during SW sleep—and identified a role for a pair of midbrain pre-isthmic nuclei, which were until now not known to be involved in sleep.

## Claustrum activity during REM$_P$ sleep

Recordings were made from the claustrum (mostly laterally) (Fig. 1a) or the adjacent anterior DVR, using silicon probes (Methods). During sleep, local field potential (LFP) activity alternated regularly between SW and REM$_P$ episodes (Fig. 1b, top), with each full sleep cycle lasting 1.5–2.5 min. SW sleep was characterized by the irregular generation of sharp-wave ripples, about once per second on average, as described previously[19,21].

REM$_P$ activity in the claustrum consisted mainly of sharp downward (negative) extracellular potentials, occurring at irregular intervals (Fig. 1b, bottom left). Those events (which we call SNs, for sharp and negative) were quite consistent in shape but variable in both amplitude and inter-event interval (IEI; Fig. 1b, bottom). IEIs, time-to-trough and

[1]Max Planck Institute for Brain Research, Frankfurt, Germany. [2]School of Life Sciences, Technical University of Munich, Freising, Germany. ✉e-mail: lorenz.fenk@brain.mpg.de; gilles.laurent@brain.mpg.de

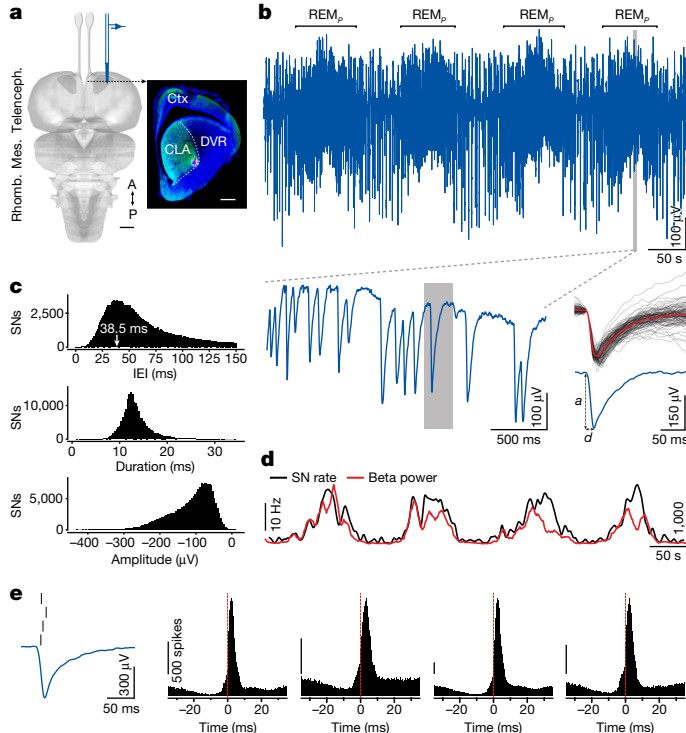

**Fig. 1 | REM$_P$ activity is characterized by sharp negative field-potential events. a**, Dorsal view of the *Pogona* brain (Mes., mesencephalon; Rhomb., rhombencephalon; Telenceph., telencephalon; A, anterior; P, posterior), with recording site (CLA, claustrum). Inset, coronal section showing the claustrum (within dashed line; green fluorescence, hippocalcin) and the electrode tip position in red (DiI fluorescence). Ctx, cortex. Scale bars, 1 mm (left); 500 μm (right). **b**, Top, LFP recording from site in **a**, during around 8 min of sleep. Epochs between REM$_P$ episodes correspond to SW sleep, characterized by sharp-wave ripples[21]. Bottom left, expanded trace within shaded window at top (fourth REM$_P$ episode). Note the single sharp negative extracellular potential (SN), expanded on the right. Bottom right, blue: trace shaded on the left (*a*, amplitude and *d*, duration of falling phase; see **c**); grey: 100 superimposed SNs and their average (red). **c**, Statistics of IEI, duration (as in **b**) and amplitude across 190,578 SNs. IEI distribution truncated at 150 ms. **d**, Instantaneous rate of SN production superimposed on power in beta band (same epoch as in **b**). **e**, Extracellular SN potentials correspond to the production of phasic and synchronized firing in claustrum units. Left, SN and four sorted single units. Right, Histograms, distributions of spike times in the four units, relative to the time of peak |d$V$/d$t$| of the SNs (red line) ($n$ = 100,632 events). Probability of these units producing at least one spike: 14–43%; probability of these units producing more than one spike: 0.3–3%. The small dip preceding the firing peak is likely to reflect the effects of down states that usually surround SNs, combined with those of the SN interval distribution during REM$_P$. Calibration bars represent 500 spikes.

amplitude distributions over about 190,000 events are shown in Fig. 1c for one lizard (from 9 h of one night). The IEI distribution was skewed with a mode at about 40 ms (median: 60.2 ms; [25th, 75th] percentiles: [39.8, 110.5] ms). Previous results[19,21] showed that REM$_P$ (in the DVR and claustrum) is dominated by LFP power in the 20-Hz (beta) band. The instantaneous power in the beta band (measured in a scrolling 10-s window) and the SN rate (Fig. 1d) were indeed well correlated, consistent with the mode of the IEI distribution. REM$_P$ activity in the claustrum is thus dominated by SNs and their interval statistics (3 animals, 9 nights; Extended Data Fig. 1a–d).

Single units isolated from claustrum recordings (Fig. 1e, left) typically fired 0–2 action potentials per SN, aligned to the SN's descending phase. Given the short duration of that phase, the action potentials of different units occurred within a few milliseconds of one another at

most (Fig. 1e and Extended Data Fig. 1e). SN waveforms are therefore extracellular potentials that reflect a net-depolarizing current in claustrum neurons; this, in turn, probably underlies the synchronized firing of the depolarized units. We will show that this phasic field potential must result from input to the claustrum, and not solely from intrinsic and coordinated properties of claustral units themselves.

## Bilateral coordination during sleep

We next recorded simultaneously from the left and the right claustra to examine the bilateral coordination of sleep-related activities. (Whereas a corpus callosum exists only in placental mammals, reptiles have several forebrain commissures[24].) The regular cycling of SW and REM$_P$ was precisely synchronized across the two sides (Fig. 2a and Extended Data Fig. 2a), but the fine coordination of the two claustra differed greatly between the two phases. During SW, the sharp-wave ripples—characteristic of this phase—were not synchronized bilaterally, and their amplitudes did not covary (Fig. 2b and Extended Data Fig. 2b,c), consistent with the independent generation of sharp-wave ripples in each claustrum[21] and the absence of reported contralateral projections between claustra in mammals[25] and in *Pogona*[21].

During REM$_P$, however, SNs in each claustrum were precisely mirrored (in timing and amplitude) on the opposite side and in sweeps from one claustrum triggered on SNs produced in the other (Fig. 2c). Over hundreds of thousands of SNs analysed, this tight correspondence between left and right sides was observed in 85 to 90% of the clearly detectable events (95–100 percentiles of SN amplitudes).

Although precisely coordinated, SNs from either side were not simultaneous, but offset by a delay of about 20 ms (median: 19.3 ms; [25th, 75th] percentiles: [15.4, 24.5] ms) with one side or the other leading (Fig. 2c). This delay was consistent throughout the night, across nights and across lizards (12 nights, 7 animals; Fig. 2d). SNs on both sides showed correlated amplitude variations (Fig. 2c), suggesting a common cause. The leading side would switch from one side to the other (Fig. 2c,d), but typically not on an event-by-event basis, as shown in a scrolling cross-correlation between left and right LFPs (Fig. 2e): switches occurred between REM$_P$ episodes (as here between the first and second), or once (occasionally twice) within single REM$_P$ episodes (as here during the second, fourth or sixth). The full sleep period from which this segment is shown is in Extended Data Fig. 3a.

The leading side at a given moment could usually be predicted by comparing the left- and right-claustrum LFPs: the side with dominant beta-band power tended to lead. In the second, fourth and sixth REM$_P$ episodes (Fig. 2e), for example, the crossing of the power curves corresponds to a leading-side switch. In the first and seventh, by contrast, one side dominated and led throughout. This correspondence between leadership and LFP power dominance was corroborated by an analysis of individual SN pairs: the SN with the larger amplitude generally led its contralateral counterpart (Fig. 2f).

The combination of matching bilateral pairs of waveforms, fixed positive or negative lags with the same absolute value and temporal lead of the stronger side suggested the possibility of competition between the two claustra, in which the stronger side at a given moment imposes its output on the weaker one, with a delay. By this hypothesis, leadership would depend on instantaneous variations of the relative 'strengths' of activity on the two sides, and the delay could be due to signal propagation and synaptic transfer in mirror-symmetric circuits. The suppression of the weaker side's own output—and its replacement by the stronger side's—suggested a winner-take-all type of bilateral competition.

## Interhemispheric competition during REM

To understand the nature of this hypothetical competition, we examined the transitions between leading sides. Two successive REM$_P$ episodes are shown in Fig. 3a, together with three short segments of

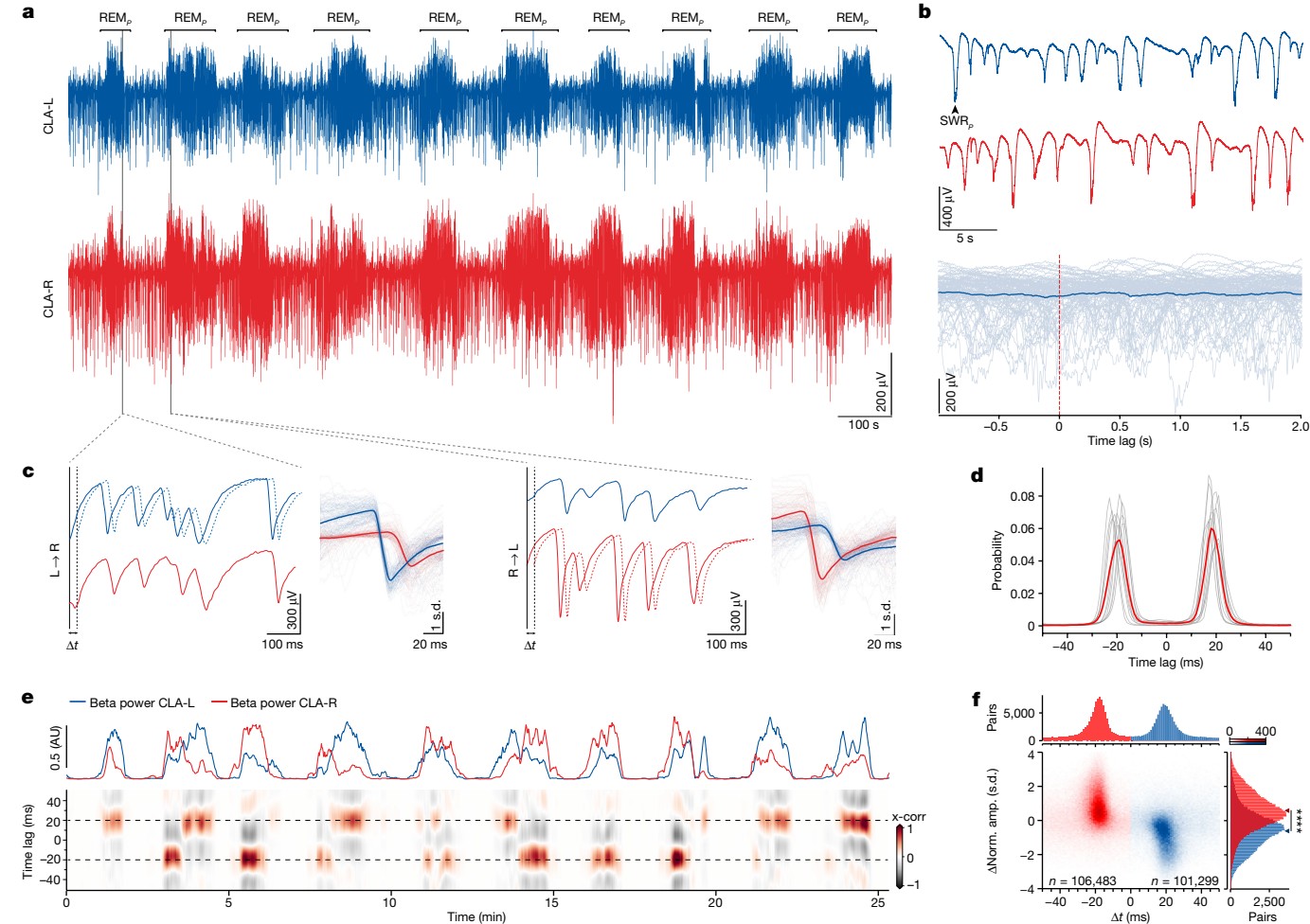

**Fig. 2 | REM_P SNs are tightly coordinated between left and right claustra, but SW sharp-wave ripples are not. a**, Paired recording from left (L, blue) and right (R, red) claustra, showing bilateral correspondence of REM_P and SW episodes. **b**, Top, absence of coordination between sharp-wave ripples (SWR_P, for *Pogona* SWR) in left and right claustra. Bottom, superimposed sweeps from one side (grey) and their average (blue), triggered from 100 single SWR_Ps from the other side (t = 0). **c**, Expanded epochs from **a** during two successive REM_P episodes. Note the similarity between left and right traces and the small time-shift between them (filled versus dashed lines). Right, superimposed sweeps of one side triggered on the other (thin lines) and their average over 100 events (thick line). Note the very tight locking of the two traces. Note also that the leading side can switch from one sleep cycle to the next. **d**, Distributions of peak-correlation lags between left and right sides, showing two symmetric

peaks at about ±20 ms (12 nights, 7 animals). Red, average. Data constructed over 9 h of sleep, containing 220,000–260,000 SNs per animal. **e**, Sliding cross-correlation (x-corr) between left and right traces (bottom) aligned with beta-band power on left and right sides. Positive lag, left side leads. Note that the side in which the beta power is larger leads, and that the leading side occasionally switches, even within the same REM_P cycle. Epochs when the beta-band power is low correspond to SW. AU, arbitrary units. **f**, Differences between side-normalized amplitudes (Norm. amp.) of around 208,000 bilateral pairs of SNs versus the time lag between them (abscissa), and marginal distributions. Colour indicates leadership (lag sign). Note that SNs on the leading side have higher amplitudes than their contralateral partner. Mann–Whitney *U*-test, *U* = 2312573585.5, ****P = 0.0. Arrowheads indicate means.

the left and right LFP traces from the second episode: in the blue frame, left power dominates and the SNs on the left lead; in the red frame, the converse happens; in the yellow frame, leadership is not settled. The yellow segment corresponds precisely to the time when the left and right beta-band powers are balanced (top traces).

By scoring an instantaneous-leadership tendency of one side versus the other (Fig. 3b and Methods) and applying it to nine hours of sleep (around 208,000 SNs), we reveal high-density domains that correspond to identifiable states: in grey is SW, when the two sides are not correlated (see Fig. 2b); in red and blue are states in which one side leads the other (right or left, respectively); and in yellow is a state in which dominance switches ('competition'). The trajectory shown in black (Fig. 3b) corresponds to the first REM_P episode in Fig. 3a; it remains on the red side from one SW state to the next. The cyan trajectory corresponds to the second REM_P episode, in which dominance starts on the left side, enters the unsettled state (yellow) and finishes on the right side before

moving back to SW. Over all recordings, the competitive (yellow) state was the least common of the three REM_P states—for example, over one night's 210 REM_P episodes, plotted in Fig. 3c, the average REM_P episode consisted of 22 s with left dominating, 26 s with right dominating and 4 s as unsettled. When one side dominated, it tended to do so over many successive SNs. Thus, if the left and right sides do compete for dominance, they must do so with dynamics slower than individual SNs, suggesting that the drive causing left–right competition has different mechanisms than those that underlie the production of SNs.

The fraction of REM_P time spent in each one of these three states (left dominating, right dominating or competition) is plotted for one night (Fig. 3c). This plot suggests that nearby REM_P episodes are not entirely independent of one another; a temporal correlation exists such that the fraction of time that either side dominated waxed and waned throughout the night (Extended Data Fig. 4). Moreover, if one side led for the longest in a REM_P cycle, it tended to do so over many

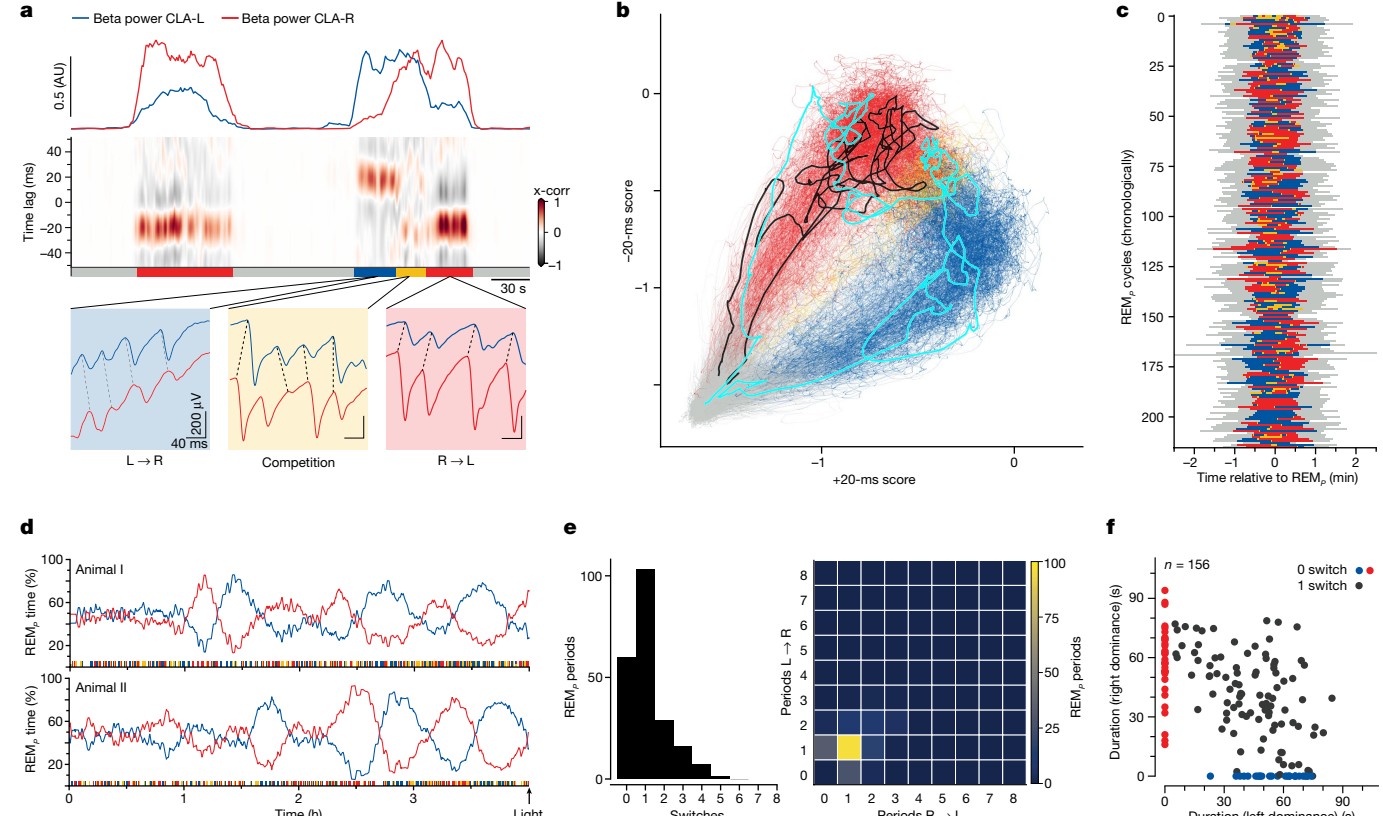

**Fig. 3 | Dominance switching obeys slow dynamics. a**, Detailed temporal relationship between left and right SNs during two $REM_P$ cycles. Blue shading, left leads. Red shading, right leads. Yellow shading, leading side not established; corresponds to time when beta-band powers are balanced (top traces). **b**, Dynamics of bilateral correlations plotted in a space defined by an instantaneous metric of temporal leadership (see Methods). Along $x$, left leads; along $y$, right leads. Each line represents one $REM_P$ cycle, starting from the grey area ($-1.5$, $-1.5$), which corresponds to SW sleep. Details in the main text. **c**, Plot of the fraction of time spent in each state (colours as in **a**,**b**) over 220 successive sleep cycles. **d**, Fraction of $REM_P$ time that each side spends leading in the four successive sleep cycles (sometimes more than ten) in the last two to three hours of the night (Fig. 3d). This long-lasting side dominance at the end of the night thus indicates the existence of a slow competitive process between the two sides of the brain, on a timescale longer than that of the sleep cycle, and increasingly long as the night proceeds.

final hours of sleep (two animals). Note the long cycles, corresponding to many (more than ten) sleep cycles, of one-sided dominance tendency. **e**, Frequency distribution of leading-side switches per $REM_P$ episode. Left, distribution of total numbers of switches. Right, distribution of number of periods of dominance per $REM_P$ cycle. **f**, Duration of $REM_P$ episodes, separating those with one switch (black) from those with none (colours). When a $REM_P$ episode contains one switch, the mean total duration of the episode (value in $x$ plus corresponding value in $y$ for all black dots) approximates 90 s on average versus 60 s on average for episodes with no switch.

The great majority of $REM_P$ episodes contained either 0 or 1 dominance switch (Fig. 3e). $REM_P$ episodes with no switch lasted about 60 s (mean, 57 s; s.d., 16.6 s; $n = 59$), against 90 s (mean, 88.5 s; s.d., 19.6 s; $n = 103$) for those with one switch. In $REM_P$ episodes with one switch, the durations of each sub-period of dominance (left-dominant and right-dominant) self-adjusted such that the duration of the $REM_P$ period remained more or less constant (diagonal trend for black points, Fig. 3f).

## A midbrain nucleus active during $REM_P$

Guided by retrograde tracing from the claustrum, we searched for circuits that might underlie the left–right claustrum competition during $REM_P$. We found no evidence for direct connections between the two claustra, and despite anatomical evidence for ipsi- and contralateral projections from the amygdala and cortex to the claustrum (Extended Data Fig. 5a,b), lesions in or ablation of either had no effect on the bilateral switching of $REM_P$ claustrum activities (Extended Data Fig. 5c–f). We next examined areas at the mid-to-hindbrain junction, given their known involvement in $REM_P$ control in mammals[1,3,17]. Recordings were

obtained after traversing part of the optic tectum (green; Fig. 4a). We detected, from mesencephalic recording sites ventral to the caudal end of the optic tectum (Fig. 4b–e), $REM_P$ sleep activity that matched precisely activity recorded in the ipsilateral claustrum. These recordings were from a nucleus that contained large, sparsely distributed GABAergic (*Vgat*[+]) and parvalbumin-positive (*Pvalb*[+]) somata at the edge of the tegmentum, next to a tecto-thalamic fibre tract (Fig. 4b–d and Extended Data Fig. 6). We suggest that this is likely to be the homologue of the nucleus isthmi pars magnocellularis (Imc) in avians, in which it has been intensively studied[26–29].

Simultaneous recordings were made from both claustra (only the left shown) and the left Imc (Fig. 4e and Extended Data Fig. 7a–c). We noted a marked correspondence between the two (Fig. 4e–g); SNs characteristic of $REM_P$ in the claustrum aligned precisely with the Imc waveforms, which themselves corresponded to brisk firing bursts of Imc neurons (Fig. 4h). Imc field potentials led those in the claustrum by 30 ms ipsilaterally (median: $-30$ ms; [25th, 75th] percentiles: [$-33.0$, $-27.0$] ms) or 50 ms contralaterally (median: $-45$ ms; [25th, 75th] percentiles: [$-51$, $-26$] ms) (Fig. 4g,i and Extended Data Fig. 7b,c), a difference consistent with the 20 ms delay between the left and the right claustra (Fig. 2). This suggested that each claustrum receives excitatory synaptic drive coupled to Imc activity in the midbrain, and that the apparent competition between claustra (Figs. 2 and 3) might instead be the expression of bilateral competition in the midbrain.

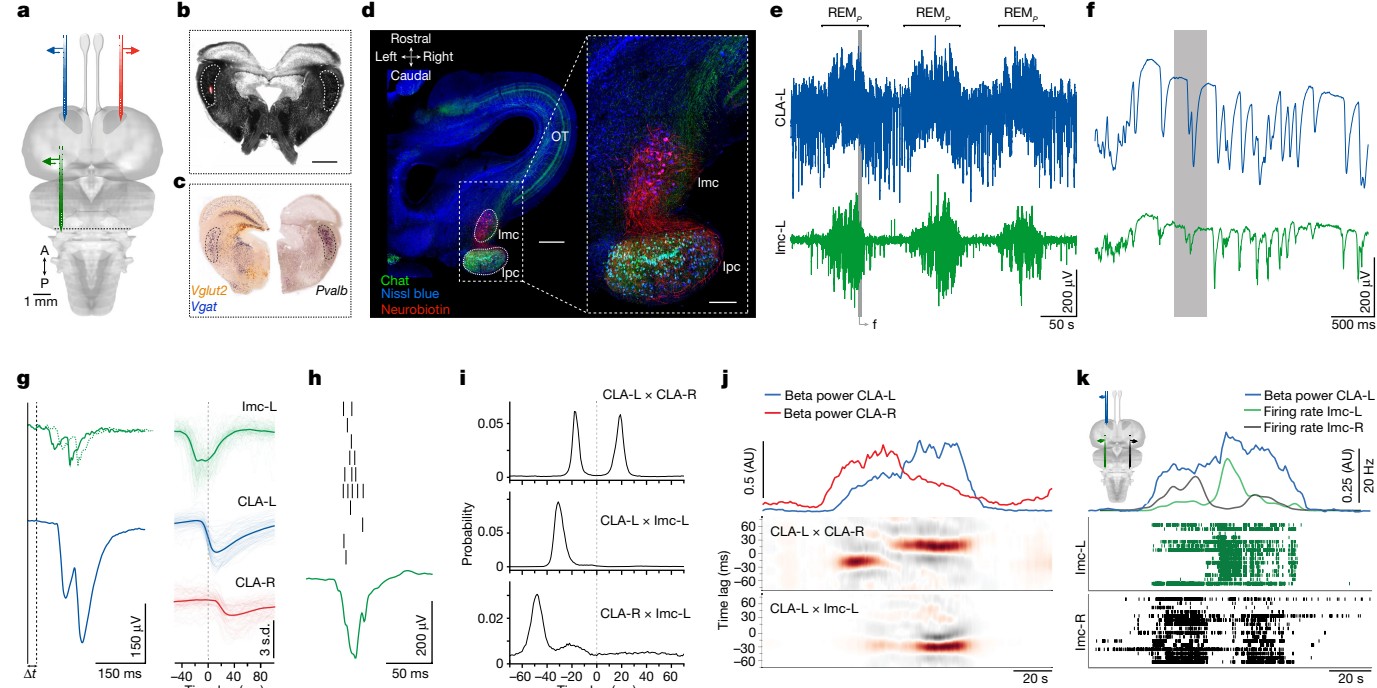

**Fig. 4 | A mesencephalic nucleus coactive with the claustrum during REM_P.**
**a**, Position of three recording probes (Neuropixels): red and blue in the claustrum; green in the posterior midbrain. **b**, Transverse section at the posterior edge of the midbrain (dark-field image). The probe-tip position is visible in red (DiI fluorescence). Scale bar, 250 μm. **c**, Immunostaining of midbrain sections at levels corresponding to that in **b**. Note large-celled GABAergic (*Vgat*⁺) and *Pvalb*⁺ nucleus (Imc). **d**, Fluorescent Nissl stain (blue) of a horizontal section of the *Pogona* midbrain (only the right hemisphere is shown). Note the Imc and its cholinergic partner, the nucleus Ipc. A neurobiotin injection labelled large cell bodies characteristic of the Imc (red). OT, optic tectum. Scale bars, 500 μm (left); 200 μm (right). See also Extended Data Fig. 6. **e**, Recordings from ipsilateral claustrum (blue) and midbrain Imc (green) during three sleep cycles. Note the activity of the Imc during REM_P. **f**, Magnification of a small section of the first REM_P cycle in **e** (LFP filtered, 100 Hz). Note the very close correspondence of the two traces. **g**, Left, further magnification of a section of the grey shaded section of the record in **f**, revealing a time lag. Right, superimposed sweeps of ipsilateral Imc (green) and contralateral claustrum (red) triggered on 100 claustrum SNs (blue), and their averages (thick lines). **h**, Each Imc negative potential coincides with a short burst of spiking activity in Imc neurons. **i**, Distributions of peak-correlation lags between left and right claustra (top); between left claustrum and left Imc (middle); and between left claustrum and right Imc (bottom). **j**, Activity of the Imc during a REM_P cycle is strongest when the ipsilateral claustrum leads. Middle, cross-correlation between left and right claustra. Bottom, cross-correlation between right claustrum and left Imc. **k**, Imc units on both sides of the midbrain show antagonistic activity during REM_P. Left claustrum activity (beta power; blue) is strongest when the ipsilateral Imc is active (green).

Indeed, although the Imc was clearly active during REM_P, its activity on a given side was strongest when the ipsilateral claustrum was also dominant (that is, phase-leading) (Fig. 4j). This suggested that the two Imcs compete with one another, and that claustrum dominance patterns reflect the results of Imc competition.

We recorded simultaneously from both Imcs (Fig. 4k and Extended Data Fig. 7d–j); sorted units from each side formed alternating bursts (two transitions in this REM_P cycle; Fig. 4k; see also Extended Data Fig. 7g), the timing of which matched power changes in the claustrum. As predicted, the instantaneous power in the claustrum increased when the ipsilateral Imc was dominant. These results suggest that inter-claustral competition in fact results from bilateral competition within the midbrain, in which the Imc has a key role.

## The Imc is required for bilateral competition

To test this hypothesis, we lesioned the Imc unilaterally using ibotenic acid (IBA) (Fig. 5a,b and Extended Data Fig. 8b) (*n* = 3 animals). The effects of these lesions were clear: the claustrum on the side of the lesioned Imc lost its ability to lead or dominate (Fig. 5c), and all REM_P episodes were now dominated by the intact side, contralateral to the lesion, as seen in two hours of one night (Fig. 5c; see full 9 h in Extended Data Fig. 8a) and in the now unimodal claustrum phase-lag distributions (Fig. 5d) (3 animals, 8 nights). Correspondingly, the phase-leading

claustrum (contralateral to the Imc lesions) was also the one with dominating beta power (blue; Extended Data Fig. 8a). Notably, lizards with unilateral Imc lesions also exhibited shorter REM_P episodes (mean duration in control animals: 65 s, *n* = 12, versus 33 s in lesioned animals, *n* = 8; *P* = 0.000028, *t* = −6.992; Welch's two-sided *t*-test) and an excess of SW, resulting in less time spent in REM_P sleep overall (mean REM_P duration: 4:40 h over 9 h of sleep in control animals, versus 1:37 h over the same duration in lesioned animals; *P* = 0.000062, *t* = −7.204), suggesting that the Imc also has a role in the SW–REM_P transition.

## Discussion

We have described competitive dynamics between the two sides of the brain, which are detectable in the claustrum during REM_P but not during SW sleep. Claustrum activity during REM_P sleep consisted of very brief synchronized firing bursts coincident with sharp negative LFP events (SNs). Contrary to sharp-wave ripples during SW sleep, SNs were tightly correlated across the two claustra, with precise time delays and rapid switches of leadership correlated with reversals of amplitude dominance, suggesting a winner-take-all type of competition between the two claustra. This competition, however, did not have a telencephalic origin: the dominance of either claustrum required the integrity of the ipsilateral member of a bilateral pair of GABAergic mesencephalic nuclei (Imc). Hence, the apparent claustral competition

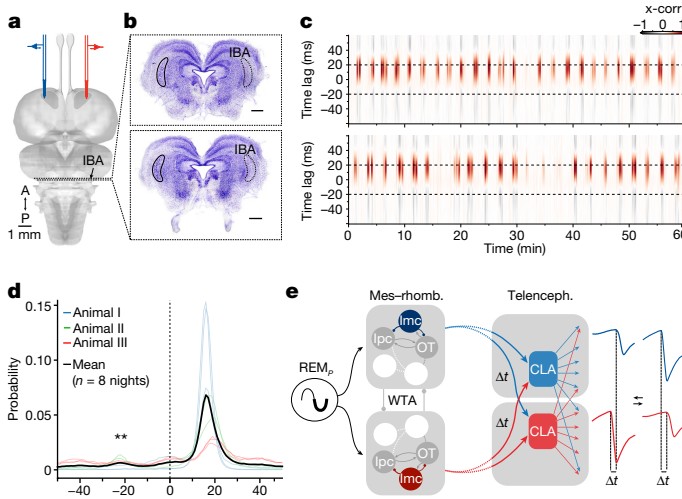

**Fig. 5 | The Imc is required for side dominance during REM_P. a**, Claustrum recording sites (red and blue) and mesencephalic IBA injection site. **b**, Assessment of unilateral lesion after experiment (Nissl stain); only a few Imc neurons remain on the injected side. Scale bars, 500 μm. **c**, Two one-hour sleep segments in the lizard in **b**, plotting the cross-correlation of the left and right claustra. Note that only the claustrum on the side of the intact Imc leads. Full night recording in Extended Data Fig. 8. **d**, Distributions of peak-correlation lags between left and right (three operated animals, unilateral Imc lesion, eight nights, mean in black) showing unimodal distribution (compare with Fig. 2d). One-sided Wilcoxon signed-rank test, $W = 36$, $**P = 0.00390625$. **e**, Schematic of the proposed functional circuits that underlie the apparent inter-claustrum competition during REM_P sleep. Mes–rhomb., mesencephalon–rhombencephalon; WTA, winner-take-all. Although the Imc is key to the left–right competition (**a**–**d**), it does not project to the other side of the midbrain, or to either claustrum in the forebrain. These projections and interactions thus rest on other relays, which have yet to be identified. The dashed lines between mid- and forebrains indicate the presence of putative relays (such as, possibly, the thalamic nucleus rotundus).

during REM_P sleep is the expression of competition in the midbrain, transferred bilaterally to the two claustra. The proposed functional circuits that underlie this relationship are schematized in Fig. 5e: in this model, the two sides of the midbrain compete during REM_P, and side dominance depends on Imc integrity (Figs. 4 and 5). Each side of the midbrain projects to both claustra (possibly, but not necessarily or exclusively, via the thalamic nucleus rotundus[30,31]) such that, as one side dominates, left and right claustrum LFPs covary (owing to their common source) but with a time lag (owing to a longer decussating path). The smaller SN amplitude on the contralateral (lagging) side also suggests a lesser gain or reliability of the midline-crossing pathway. During the brief transitions when no midbrain side dominates, the left and right claustra are coactive, which results in disordered SN patterns in both claustra (Fig. 3a).

The Imc is part of a complex of cholinergic, glutamatergic and GABAergic isthmic nuclei that has been studied mainly in birds[26,28,29,32] and is implicated in bottom-up attention and gaze control. The Imc, which is embryologically a pre-isthmic nucleus[33], mediates a type of broad lateral inhibition of excitatory neurons in both the ipsilateral optic tectum and the nucleus isthmi pars parvocellularis (Ipc)—a companion cholinergic–glutamatergic nucleus that forms a positive feedback loop with the ipsilateral tectum[27,29,31,32,34]. When two stimuli fall onto one retina, these circuits undergo a winner-take-all-type competitive interaction such that responses to the stronger stimulus are selected, and attention and gaze are directed towards it, rather than to a weighted mean of the two stimuli. The Imc, by virtue of its heterotypic connectivity with the optic tectum and the Ipc, underlies this competition[27,28]. This work was at first concerned with the selection of competing visual stimuli

falling on the same retina, but experiments have now established that competition also takes place between contralateral stimuli, and even between sensory modalities[35], although the underlying circuits are not well understood. Likely homologues of these avian (reptilian) isthmic nuclei are found in fish[36–38], amphibians[39] and non-avian reptiles[40,41], as well as in mammals (parabigeminal nucleus), in which a role in visual attention has been hypothesized[42].

The isthmo–claustral connection uncovered here is notable in part because some of the claustrum's hypothesized functions are attentional[43,44]. But our results raise many new questions. First, what pathways underlie the bilateral mesencephalic competition (Fig. 5e)? Although the Imc is inhibitory (Fig. 4c) and necessary for this competition (Fig. 5), we are not aware of contralateral Imc projections. Second, what pathways link Imc, a GABAergic nucleus, to the activation of both claustra? The high correlation between isthmic and claustral LFPs during REM_P suggests high-gain excitatory pathways between those areas, or drive from a common source, the dominance of which depends on the integrity of the ipsilateral Imc. Third, what is the functional importance of the tight correlation and ±20-ms delays between the claustra during REM_P? Although there is no evidence for direct projections between the claustra, bilateral claustral projections to the telencephalon exist[25,44], such that convergence on common targets is plausible. If so, the delays might, in conjunction with the activation of spike-timing-dependent plasticity rules, have a role in synaptic homeostasis[16,45,46] or plasticity during REM_P. These hypotheses are especially intriguing when considering that the temporal profiles of dominance are different early and late in the night (Fig. 3), and that sharp-wave ripples are uncorrelated bilaterally during SW (Fig. 2 and Extended Data Fig. 2). Fourth, several species of birds and marine mammals show unilateral sleep[47,48]. Could the same mesencephalic circuits underlie this (slower) alternation? Fifth, side dominance is temporary and balanced between the two sides over each night. What is the cause of the recurrent transfer of side dominance (Fig. 3), presumably located upstream of the Imc? Because the duration of one-sided dominance varies between a fraction of one sleep cycle and many cycles, it probably does not originate in the circuits that cause SW–REM_P alternation. Sixth, what role does the isthmic complex have in SW–REM_P transitions? Imc lesions revealed a decrease in REM_P and a concurrent increase in SW sleep, indicating an interaction with circuits that control the ultradian sleep cycle. Finally, how are activities between the isthmi and the claustra coordinated in the awake state, given that brain activity during REM_P sleep most resembles that in the awake state[9,17]?

Hence, the claustrum—a well-described[44,49] but functionally poorly understood area of the telencephalon—seems to have a role in SW sleep in reptiles and mammals[21,50], and reflects competitive brain dynamics during REM_P sleep (this paper), with origins in the midbrain. Antiphase activity between the two hemispheres has been identified in running rodents and during REM with fast rhythms (140-Hz 'splines') in superficial retrosplenial cortex[6], but the mechanisms that underlie this relationship are unknown. The relative simplicity of the reptilian model may thus help to reveal not only some mechanisms of sleep, but also some of the functions of the claustrum and of the midbrain in sleep.

Finally, our data emphasize the value of investigating non-mammalian sleep. Comparing stages of sleep (for example, SW and REM), electrophysiological waveforms (for example, SWRs) or even brain areas (for example, the claustrum) across distant species can be difficult; indeed, the terminology of sleep originated with data from humans and cats, the phenomena to be compared are usually multiparametric and sleep is polymorphic even among mammals. Matches across species are thus often only partial. But by comparing data using exhaustive and increasingly precise approaches, hypotheses about structural homologies can now be tested more accurately, at least among vertebrates. This should in turn help us to determine which phenotypic and mechanistic attributes of sleep are common across species, how they came to diverge or converge, and eventually whether they subserve similar functions.

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

## Methods

### Animals

Lizards (*P. vitticeps*) of either sex, weighing 150–250 g, were obtained from our institute colony and selected for size, weight and health status. The lizards were housed in our state-of-the-art animal facility. All experimental procedures were approved by the relevant animal welfare authority (Regierungspräsidium Darmstadt, Germany) and conducted following the strict federal guidelines for the use and care of laboratory animals (permit numbers V54-19c20/15-F126/1005_1011 and 2006).

### Lizard surgery for chronic recordings

Twenty-four hours before surgery, lizards were administered analgesics (butorphanol, 0.5 mg kg$^{-1}$ subcutaneously; meloxicam, 0.2 mg kg$^{-1}$ subcutaneously) and antibiotics (marbofloxacin, marbocyl, 2 mg kg$^{-1}$). On the day of surgery, anaesthesia was initiated with 5% isoflurane, and maintained with isoflurane (1–4 vol%) after intubation. The lizards were placed in a stereotactic apparatus after ensuring deep anaesthesia (absence of corneal reflex). Body temperature during surgery was maintained at 30 °C using a heating pad and an oesophageal temperature probe. Heart rate was continuously monitored using a Doppler flow detector. The skin covering the skull was disinfected using 10% povidone-iodine solution (Betai-NE) before removal with a scalpel. A cranial window was made to reach the various regions of interest; the dura and arachnoid were then removed with fine forceps and scissors. The pia was carefully removed over the area of electrode insertion (dorsal cortex for claustrum and optic tectum for Imc recordings). The exposed skull was covered with a layer of ultraviolet (UV)-hardening glue, and the stripped ends of insulated stainless steel wires were secured in place subdurally with UV-hardening glue, to serve as reference and ground.

Silicon probes were mounted on a Nanodrive (Cambridge Neurotech) and secured to a stereotactic adaptor for insertion. On the day after the surgery, probes were slowly lowered into the tissue (claustrum: 0.8–1.2 mm; Imc: 2.5–3.0 mm). For Imc recordings, the probe was advanced in small steps over two to three days until the signal began to show the area-typical signature.

The brain was covered with Duragel (Cambridge Neurotech) followed by Vaseline, and the probes were secured with UV-hardening glue. After surgery, lizards were released from the stereotactic apparatus and left on a heating pad set to 30 °C until full recovery from anaesthesia.

### In vivo electrophysiology

Two to three days before surgery, lizards were habituated to a sleep arena, which was itself placed in a 3 × 3 × 3-m electromagnetic-shielded room. One hour before lights off ($t_{off}$ = 18:00/19:00, winter/summer), lizards were placed into the arena and left to sleep and behave naturally overnight. They were returned to their home terrarium 3 to 4 hours after lights on ($t_{on}$ = 06:00/07:00, winter/summer). The lizards then received food and water. Experiments were performed at a constant room temperature of around 21.5 °C.

Electrodes were either 32-channel silicon probes (NeuroNexus; 50-µm pitch, 177-µm$^2$ surface area for each site; in 2 rows of 16 contacts), or Neuropixels 1.0 probes[51]. Recordings using 32-channel probes were performed with a Cheetah Digital Lynx SX system and HS-36 headstages as previously described[21]. Signals were sampled at 32 kHz, and IronClust with manual curation was used for spike sorting (https://github.com/flatironinstitute/ironclust#readme). Neuropixels data were acquired nominally at 30 kHz using SpikeGLX software (http://billkarsh.github.io/SpikeGLX/). The true sampling frequency varied slightly with different headstages and could drift throughout a recording. Data acquired from multiple probes were synchronized by linear interpolation, using as a reference a common 1-Hz squarewave signal recorded with all probes.

Action potentials were sorted using Kilosort2 (ref. [52]; https://github.com/MouseLand/Kilosort) and the ecephys pipeline (https://github.com/jenniferColonell/ecephys_spike_sorting); clusters were curated manually in Phy (https://github.com/cortex-lab/phy).

### Lesion experiments

In preparation for Imc lesion experiments, we carefully removed the pia overlaying the optic tectum and inserted a bevelled quartz micropipette at an angle of 6–8° from the vertical axis to depths of 2,700–2,900 µm. IBA (200–350 nl; 10 µg µl$^{-1}$ in phosphate-buffered saline (PBS), pH 7.2) was injected at a rate of 50–100 nl min$^{-1}$ (UMP3, World Precision Instruments). The injection pipette was retracted 3–5 min after the end of injection.

The cortical sheets of *Pogona* overlay the lateral ventricles and can be removed using surgical scissors. This operation was preceded by removing the overlaying pia and vasculature, staunching bleeding using forceps. For surgical lesions of the amygdala, we first removed the cortical sheet and then the amygdala (together with caudal parts of the DVR), using fine forceps and scissors.

Silicon probes were positioned bilaterally above the two claustra and recordings were performed each night from one to six days after surgery. The effects of lesioning Imc could be observed 24 h after surgery and were stable thereafter. By the end of the experiment, the lizard was euthanized and its brain sectioned and Nissl-stained for histological confirmation of the lesion.

### Cross-correlations

To compute lagged cross-correlations between 2 channels (for example, Fig. 2e) the signals were first downsampled to 1 kHz, low-pass-filtered to 40 Hz and $z$-scored. Lagged cross-correlations were computed on the first derivative of the resulting time series with a sliding window of 10 s scrolled in 100-ms steps. Cross-correlations were normalized to the value of their 99.9% percentile.

To estimate the lags that best represent the relationship between two signals, we extracted distributions of peak-correlation lags (for example, Fig. 2d). We took, for every time point in the lagged cross-correlation, the lag corresponding to the maximum value, producing a time series of lags. To isolate periods of high correlation, we kept only lags corresponding to correlation values in the top 75% percentile. To avoid boundary artefacts resulting from a finite exploration of lags, we removed those at the extremes of the range (for example, −51 ms and +51 ms in Fig. 2d).

To score the instantaneous-leadership between 2 channels, we extracted their cross-correlation using the lags corresponding to the modes of the distribution of peak-correlation lags (−20 ms and +20 ms in Fig. 3b). Before extracting the cross-correlation, we processed the signals as described above and we then clipped their first derivative to be less than or equal to zero. Note that the resulting cross-correlation must be greater than or equal to zero. We then took the base-10 logarithm of the resulting cross-correlation values

$$s_{\pm 20}(t) = \log_{10}\left[\sum_{\tau=-w/2}^{+w/2} [c_0(t+\tau)]^- \times [c_1(t+\tau\pm 20)]^-\right],$$

where $w$ is the 10-s sliding window and $c_0$ and $c_1$ are the first derivatives of the 2 channels as described above. To define periods of dominance (Fig. 3), we selected linear thresholds of the −20 ms score and the +20 ms score manually. To avoid over-fragmentation resulting from the noisy crossing of these thresholds, we ignored detours that left and re-entered a single state for a short duration (less than 3 s).

### REM$_P$ detection

To define periods of REM$_P$ and SW sleep, we extracted the power of the LFP signal in the beta band (12–30 Hz) with a 10-s sliding window at 1-s steps. We considered periods with a beta power above its 15th percentile as REM$_P$ sleep periods. To avoid over-fragmentation resulting from the noisy crossing of this threshold, we ignored detours that left and re-entered the same state for a short duration (less than 15 s).

## SN detection and matching

To detect the SNs typical of claustrum REM$_P$ activity, we first low-pass-filtered the signal to 40 Hz and extracted its first and second derivatives. We then extracted peaks in the resulting time series. We considered a potential SN as a triplet of peaks: a negative second derivative peak followed by a negative first derivative peak and a positive second derivative peak. The first and last peaks correspond to the beginning and end of the downward phase of an SN; we used them to calculate amplitude and duration (Fig. 1b,c). To remove false positives from this set of potential SNs, we estimated the distribution of the noise and took only those SNs with a low probability ($P < 0.025$) on the corresponding cumulative distribution function (CDF) of negative amplitude and duration. To estimate the distribution of noise—that is, of small LFP deflections wrongly identified by our method as SNs—we multiplied the signal by −1 and repeated the same process of triplet peak detection. This was equivalent to attempting to detect sharp positive deflections, which did not exist in the signal. Consequently, any positive events detected by our triplet peak detection were the result of LFP noise. We then used those wrongly identified positive events to establish minimum thresholds of amplitude and duration on the originally detected SNs.

To match bilateral pairs of potential SNs, we scored each pair by the value of the lagged cross-correlation at their corresponding time and lag. The lagged cross-correlation was computed with a window of 100 ms (see 'Cross-correlations') and multiplied by an exponential kernel with a time constant of 10 ms. We then algorithmically detected the optimal combination of potential SN pairs that maximized the total sum of this score. Note that an SN could have only one match and that some could be left without a matching partner. Pairs of potential SNs were evaluated against the noise CDF and were accepted if at least one of the two potential SNs had a low null probability ($P < 0.05$).

The SN detection and matching process was performed in consecutive sections of 1 h duration.

## Tract tracing

For ex vivo tracing experiments (Extended Data Fig. 6), lizards were deeply anaesthetized with isoflurane, ketamine (60 mg kg$^{-1}$) and midazolam (2 mg kg$^{-1}$). After loss of the corneal reflex, the lizards were decapitated and their heads immediately submerged in ice-cooled artificial cerebrospinal fluid (aCSF) (126 mM NaCl, 3 mM KCl, 1.8 mM CaCl$_2$, 1.2 mM MgCl$_2$, 24 mM NaHCO$_3$, 0.72 mM NaH$_2$PO$_4$ and 20 mM glucose). We then perfused the brains with cooled aCSF, extracted them and carefully removed the pia above injection sites. Neurobiotin (5–10% dissolved in phosphate buffer) was delivered through glass micropipettes and iontophoresis, applying 5-µA current pulses (5 s on, 5 s off) for 2–10 min. After injection, the brains were kept submerged in aCSF at room temperature for 15–20 h to allow for transport of the tracer, before being transferred to 4% paraformaldehyde in PBS for 24–48 h at 4 °C. After fixation, the brains were immersed in 30% sucrose for at least 48 h at 4 °C. Transverse or horizontal sections were obtained using a cryostat, at a thickness of 70 µm, and neurobiotin was detected with streptavidin, Alexa Fluor 568.

## Statistics

Statistical tests were performed using the standard Python package scipy (v.1.6.2). $P$ values equal to zero indicate values too small ($< 5 \times 10^{-324}$) to be computed with a standard float64 and result from the large number of samples in our recordings.

## Data reporting

No statistical methods were used to predetermine sample size. The experiments were not randomized and the investigators were not blinded to allocation during experiments and outcome assessment.

## Reporting summary

Further information on research design is available in the Nature Portfolio Reporting Summary linked to this article.

## Data availability

Data will be made available upon reasonable request.

## Code availability

Analysis code is available at: https://brain.mpg.de/research/laurent-department/software-techniques.

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

**Acknowledgements** We thank E. Northrup, N. Vogt and G. Wexel for veterinary care; E. Joesten, T. Klappich, M. Lange and M. de Vries for reptile care; A. Arends, M. Klinkmann, J. Knop, A. Macias Pardo and C. Thum for technical assistance; T. Gallego-Flores for contributing in situ stains; S. Weiss for help in setting up Neuropixels recordings; L. Puelles for feedback on the identity and mesencephalic origin of the Imc; L. Faraggiana, J. Gjorgjieva and H. Norimoto for discussions; and D. Evans and H. Ito for their comments on the manuscript. This work was funded by the Max Planck Society (G.L.), the European Research Council under the European Union's Horizon 2020 research and innovation programme (grant agreement no. 834446) (G.L.) and the DFG (CRC1080) (G.L.). L.A.F. was supported by an EMBO long-term fellowship (ALTF 421-2017).

**Author contributions** L.A.F. and G.L. designed the project. L.A.F. performed the experiments. All authors discussed and interpreted the results. J.L.R. and L.A.F. analysed the data and prepared the figures. G.L. wrote the manuscript, with contributions from J.L.R. and L.A.F., and supervised the project.

**Funding** Open access funding provided by Max Planck Society.

**Competing interests** The authors declare no competing interests.

**Additional information**
**Correspondence and requests for materials** should be addressed to Lorenz A. Fenk or Gilles Laurent.

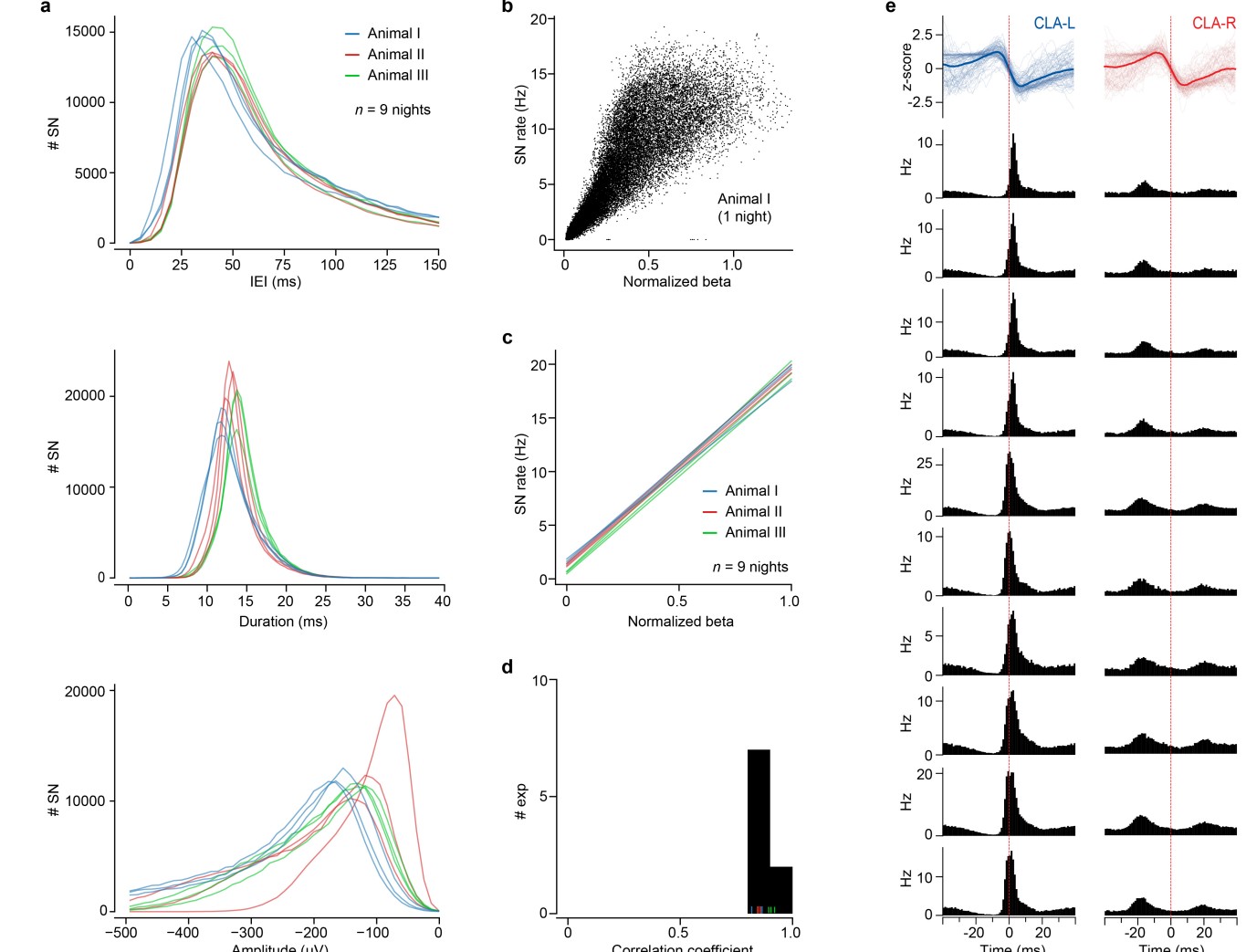

**Extended Data Fig. 1 | SN characteristics for multiple lizards and nights.**
**a**, Distributions of inter-event interval (IEI), SN duration and amplitude for 9 recordings and 3 animals. IEI distribution truncated at 150 ms. **b**, SN rate (Hz) and corresponding normalized beta power for one recording. Each dot corresponds to a sample taken every second on a sliding window of 10 s for 9 h of sleep. **c**, Linear fits to SN rate and normalized beta power as in **b** across all 9 recordings. **d**, Distribution of Pearson's r of linear fits in **c**, showing very high correlation of SN rate and beta power. **e**, Extracellular SN potentials coincide with phasic and synchronized firing in claustrum units. Data as in Fig. 1e, but data taken from a different animal, with bilateral claustrum recording.

Top: 128,453 (left) and 123,798 (right) SNs together with the mean waveforms (bold). Histograms: distributions of spike times in 10 units recorded in left claustrum, relative to the time of peak |dV/dt| of the SNs (red stippled line) detected either ipsilaterally (left) or contralaterally (right). Right column: Note peaks around −20 ms and +20 ms, corresponding to periods when the contralateral and ipsilateral side are leading, respectively. Also note that peaks are higher when the contralateral claustrum is leading (the side from which the spikes shown are recorded), consistent with greater synaptic drive and larger amplitude SNs in the dominant hemisphere (cf Fig. 2c–f).

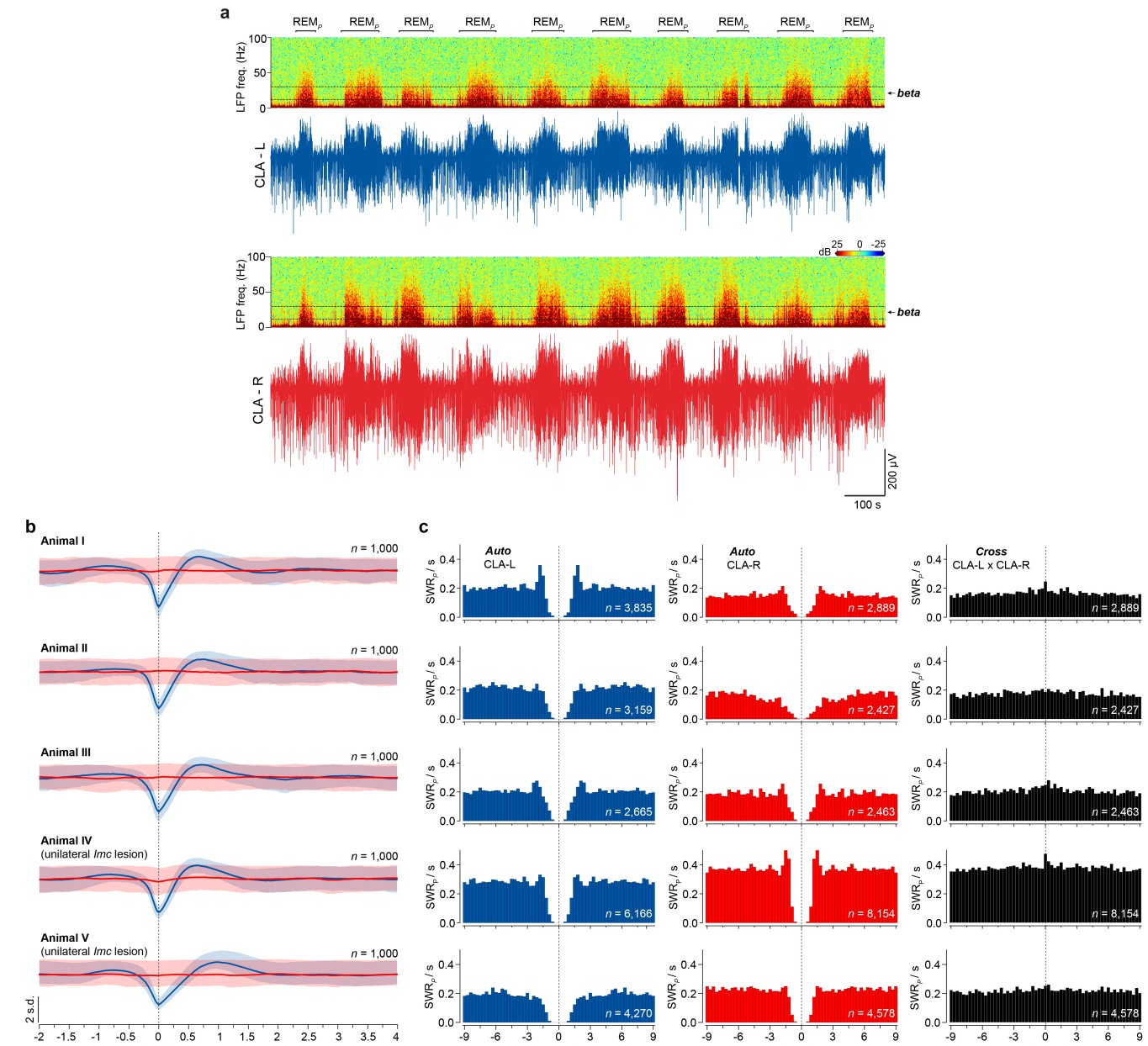

**Extended Data Fig. 2 | SWR$_P$ production in the claustrum is not coordinated on the left and right sides during SW sleep. a**, LFP traces from the paired recording shown in Fig. 2a, with their band spectrograms (0.1–100 Hz). Note beta band used to define onset and offset of REM$_P$. **b**, Averages (blue) of 1,000 SWR$_P$s randomly picked through 9 h of sleep, together with the SWR-triggered average LFP (red) on the contralateral side (shaded areas: standard deviation). For details on the Imc and its identification, see Fig. 4 and Extended Data Fig. 6. **c**, Corresponding auto- and cross-correlograms for animals I–V in b. Note flat cross-correlograms.

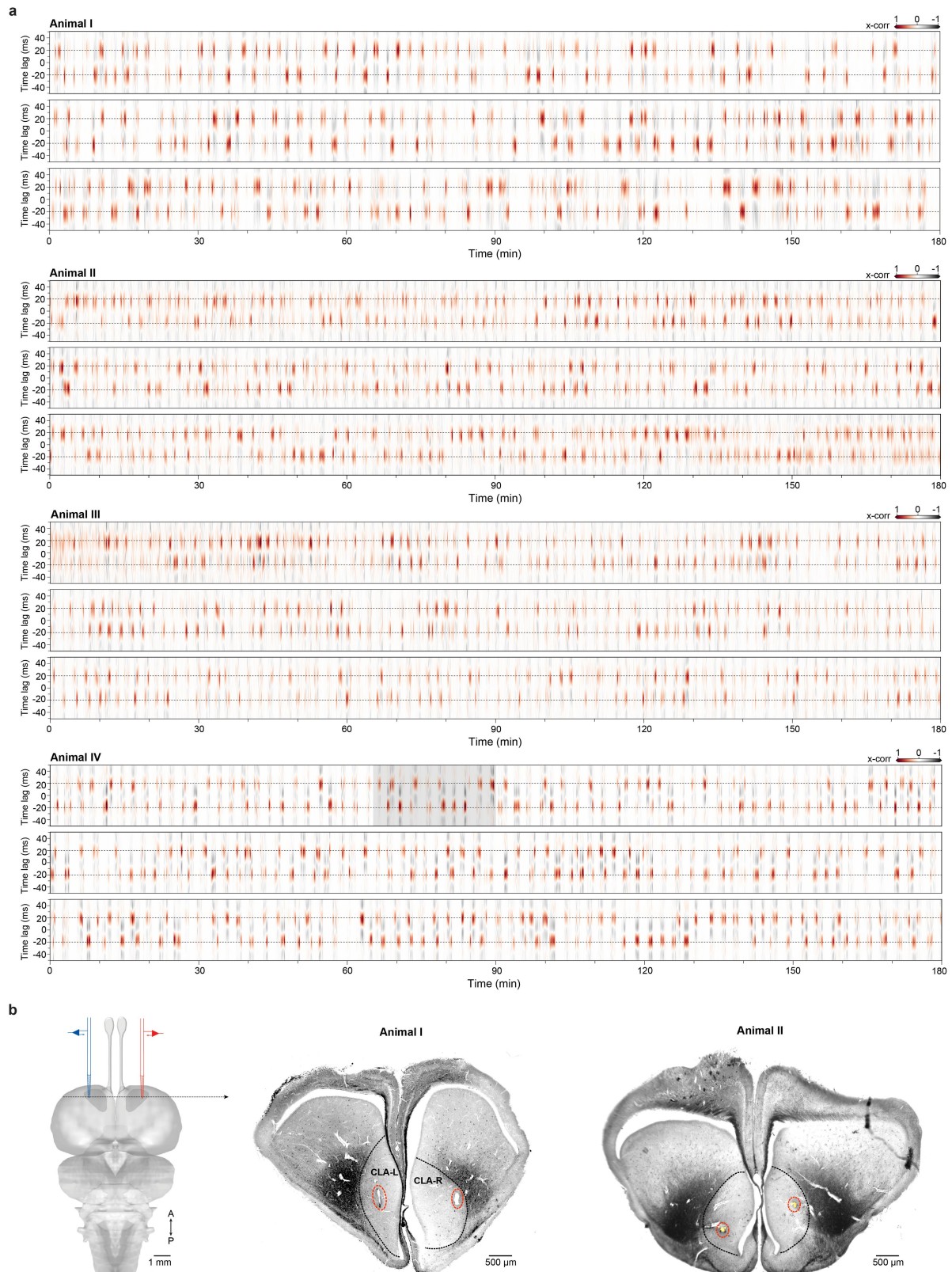

**Extended Data Fig. 3 | Cross-correlations between bilaterally recorded claustrum LFPs over nine hours of sleep. a**, Cross-correlation between left and right claustra for four animals and nine hours of sleep each. The shaded area (animal IV) corresponds to the segment shown in Fig. 2e. **b**, Recording locations in the left and right claustra, for two representative examples. Left, schematic of recording configuration. Middle and right, transverse sections through the anterior telencephalon (dark-field image), highlighting the recording sites (red stippled line), as identified by electrolytic lesions and DiI dye applied to the back of the silicon probes.

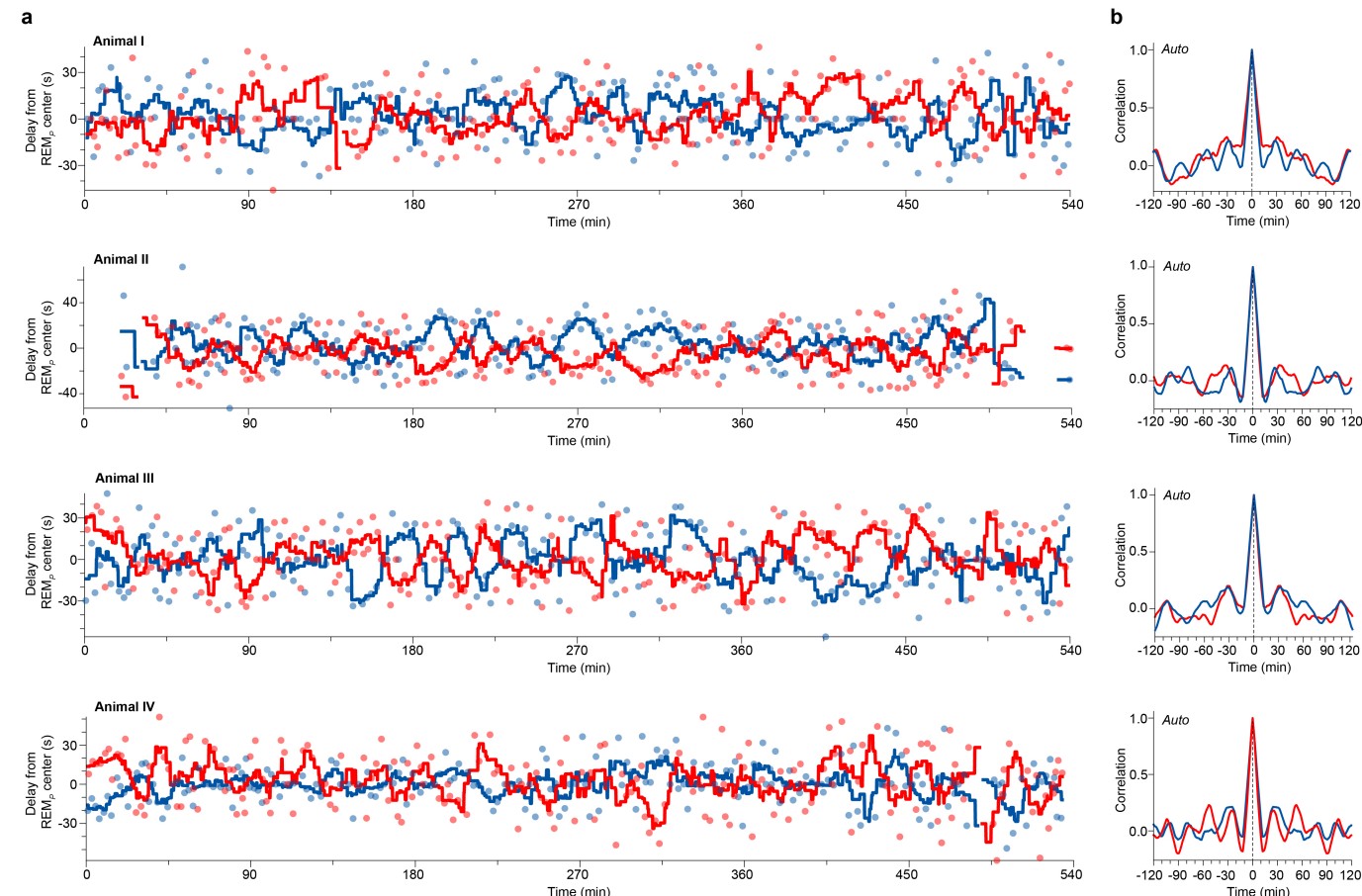

**Extended Data Fig. 4 | Dynamics of dominance switches. a**, Evolution of the periods of dominance over consecutive REM$_P$ cycles for four animals and nine hours of sleep each. Each dot represents the time of the main period of dominance for one cycle, relative to that cycle's centre (blue, left; red, right). Lines are running averages in a 12-min window. Note the order within REM$_P$ and their durations fluctuate with slow dynamics, longer than a single REM$_P$ cycle. **b**, Auto-correlations of running averages in **a**. Note that animals often display positive correlations at around 30 min.

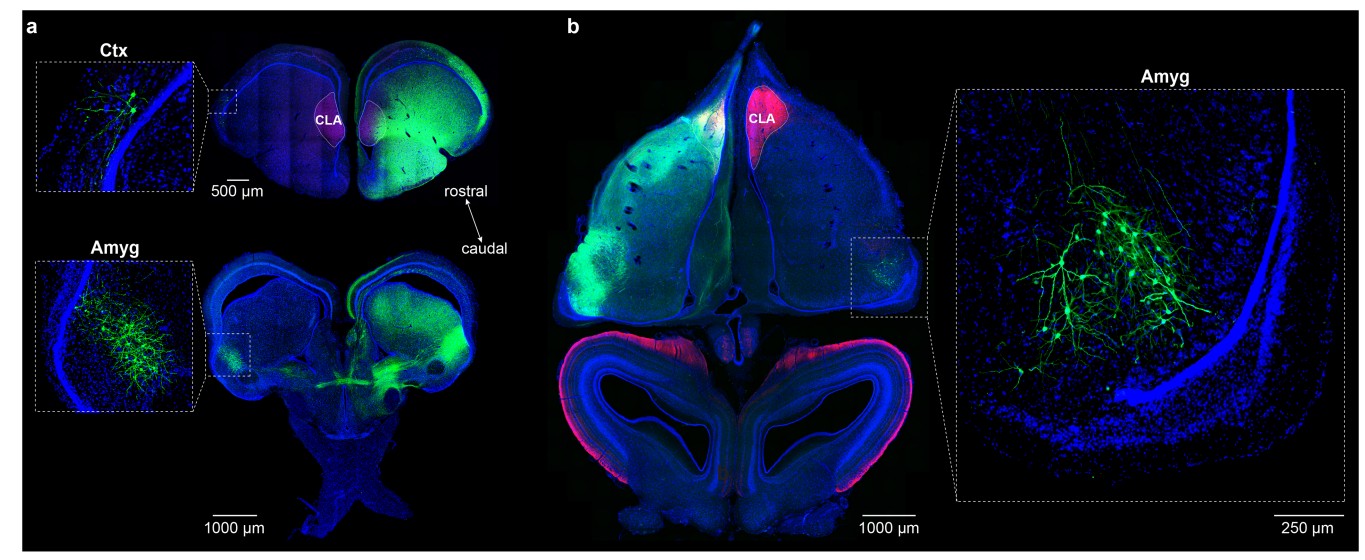

### a

Ctx

Amyg

CLA

500 µm

1000 µm

rostral

caudal

### b

CLA

Amyg

1000 µm

250 µm

### c

**Cortex Lesions**

*n* = 6 nights
(4 animals)

— unilateral (dCtx)
— bilateral
— bilateral
— bilateral
···· mean

Probability

Time (ms)

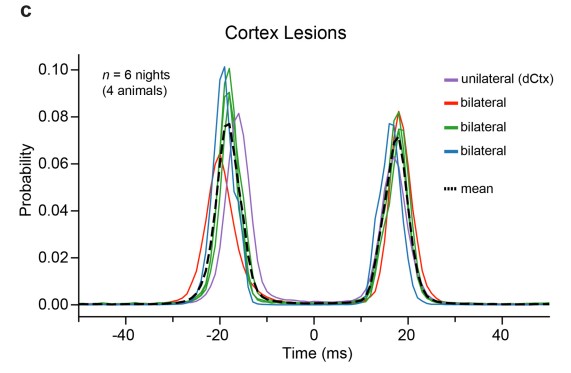

### d

**Amygdala Lesions**

*n* = 4 nights
(4 animals)

— unilateral (IBA)
— unilateral (IBA)
— unilateral (surgical)
— bilateral (surgical)

} + Lesion of Ctx,
parts of DVR,
parts of striatum

···· mean

Probability

Time (ms)

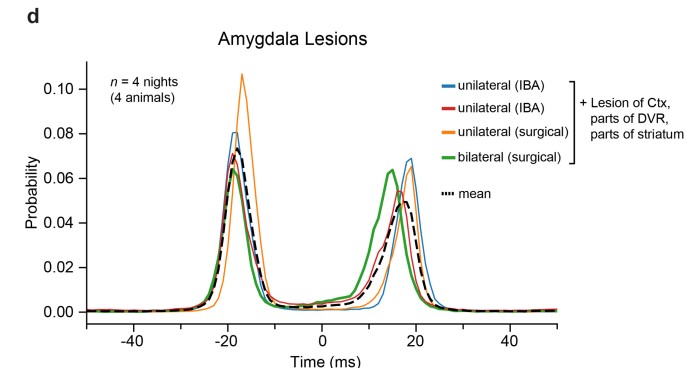

### e

Bilateral cortex lesion (surgical)

Ctx

Ctx

DVR

DVR

CLA

CLA

500 µm

500 µm

rostral

caudal

Ctx

Ctx

DVR

DVR

Sept.

Sept.

Striat.

Striat.

500 µm

500 µm

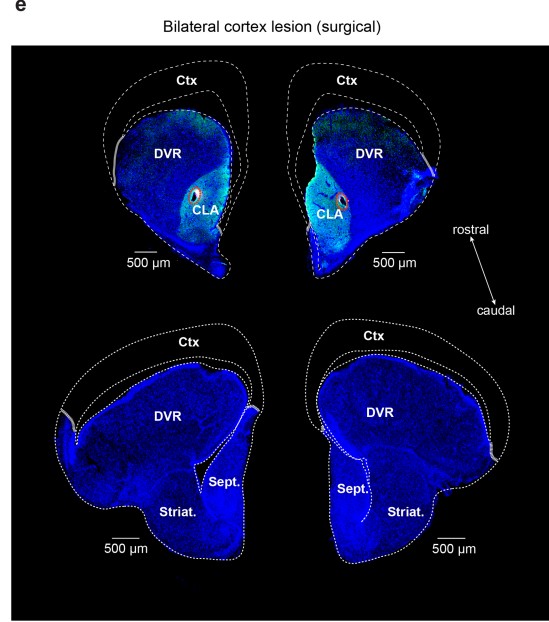

### f

Bilateral amygdala lesion (surgical)

Ctx

Ctx

DVR

DVR

500 µm

rostral

caudal

Ctx

Ctx

DVR

DVR

Amyg

Striat.

Sept.

Sept.

Striat.

Amyg

Optic tract

500 µm

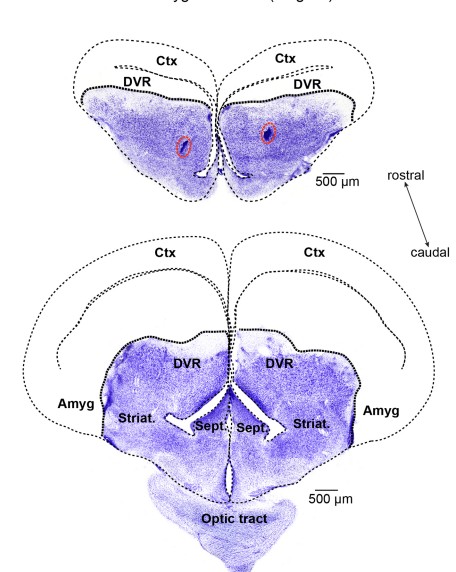

**Extended Data Fig. 5** | See next page for caption.

**Extended Data Fig. 5 | Cortex and amygdala lesions do not prevent dominance switches. a**,**b**, Identification of contralateral inputs and distribution of GFP-labelled neurons in the forebrain after injection of rAAV2-retro into the *Pogona* claustrum on one side. **a**, Transverse sections through the telencephalon, showing very sparse labelling in the contralateral cortex (top), and dense labelling of neurons in the contralateral amygdala. **b**, Horizontal section through the tel- and mesencephalon, showing contralateral input from the amygdala. Also note the absence of labelled cells in the contralateral claustrum (within stippled line; pink fluorescence: hippocalcin). **c**, Distributions of peak-correlation lags between left and right claustra for four animals with cortex lesions. **d**, Same as in **c**, but for amygdala lesions. Note that we removed the cortical sheet before injecting IBA for excitotoxic lesions, and before removing the amygdala surgically. Additional lesions were made to the adjacent DVR and parts of the striatum. None affected the inter-claustral apparent competition. **e**, Transverse sections at the level of the claustrum and more caudal telencephalon, showing the recording locations (red circles, top), and the bilateral absence of cortex along the rostro-caudal axis (top and bottom). **f**, Nissl stains of transverse sections at the level of the claustrum and caudal telencephalon, comparable to **e.** Indicated are the approximate positions of the bilaterally lesioned amygdala and other brain areas.

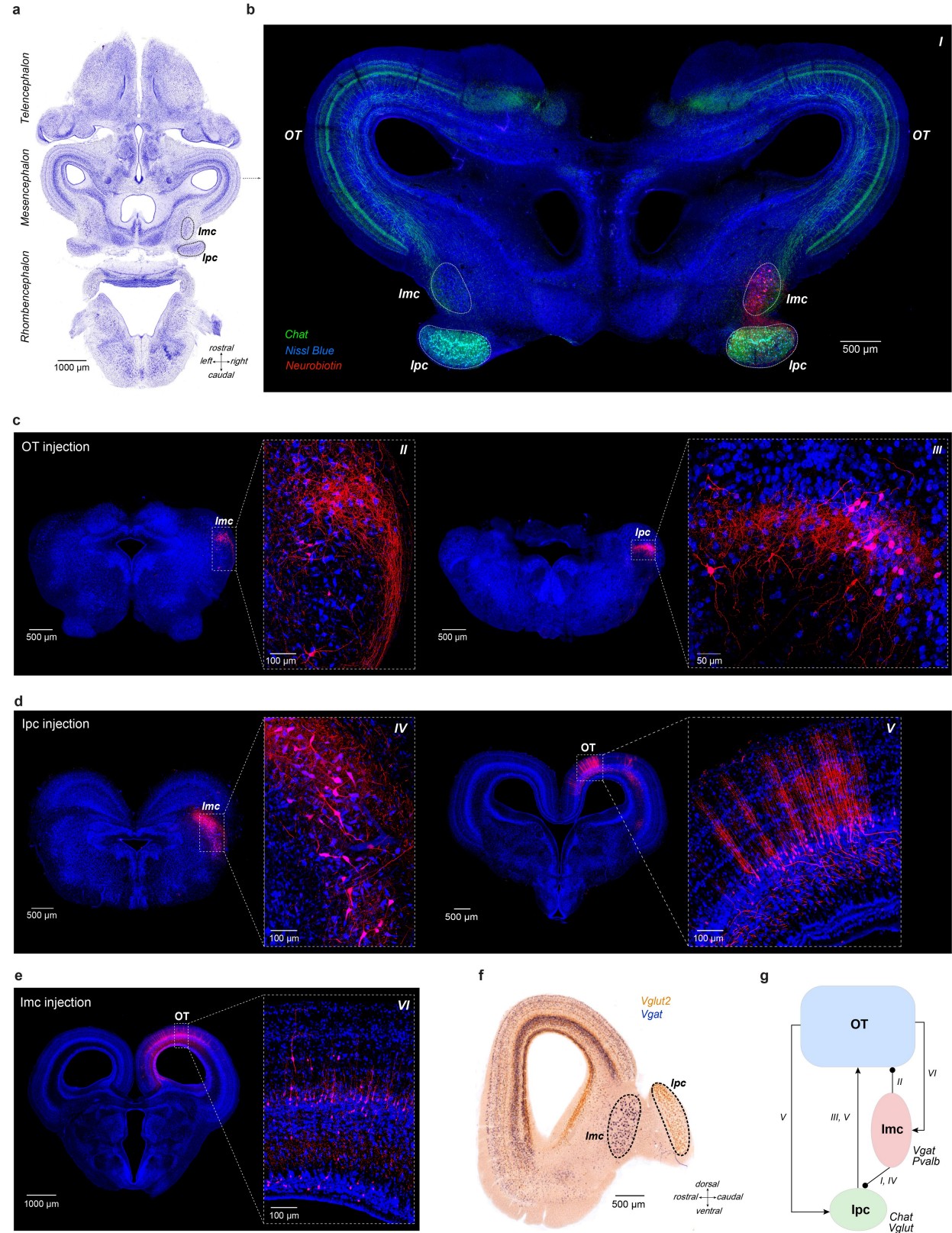

**Extended Data Fig. 6** | See next page for caption.

**Extended Data Fig. 6 | The isthmic nuclei (Imc and Ipc) and their connections.** **a**, Nissl stain of a horizontal section of the *Pogona* brain. The Imc and Ipc are located at the mid- to hindbrain junction, and are highlighted on the right hemisphere (black stippled ellipsoids). **b**, Fluorescent Nissl stain (blue) of a horizontal section through the mesencephalon, plus the Ipc (midbrain-hindbrain transition). Note the small, cholinergic cell bodies characteristic of the Ipc (green), and cholinergic fibres extending into the optic tectum (OT; superior colliculus in mammals). A neurobiotin injection labelled large cell bodies in the Imc (red), whose axons innervate the Ipc. **c**, Retrograde labelling of cell bodies in the Imc (left) and Ipc (right) after injection of neurobiotin into the OT. **d**, Retrograde labelling of cell bodies in the Imc (left) and OT (right) after injection of neurobiotin into the Ipc. Also note the fine ramifications of neural processes in the OT, probably corresponding to 'paintbrush' axon terminals of the Ipc described by previously[53], and later shown to provide focal, re-entrant amplification of retinal input to the OT. **e**, Retrograde labelling of cell bodies in the OT, after injection of neurobiotin into the Imc. **f**, Sagittal section through the lateral mesencephalon, and Ipc. Double in situ hybridization with *Vglut2* and *Vgat* probes. **g**, Summary diagram of the ipsilateral connections shown in **b**–**e**; expression of marker genes shown in **f** and Fig. 4c.

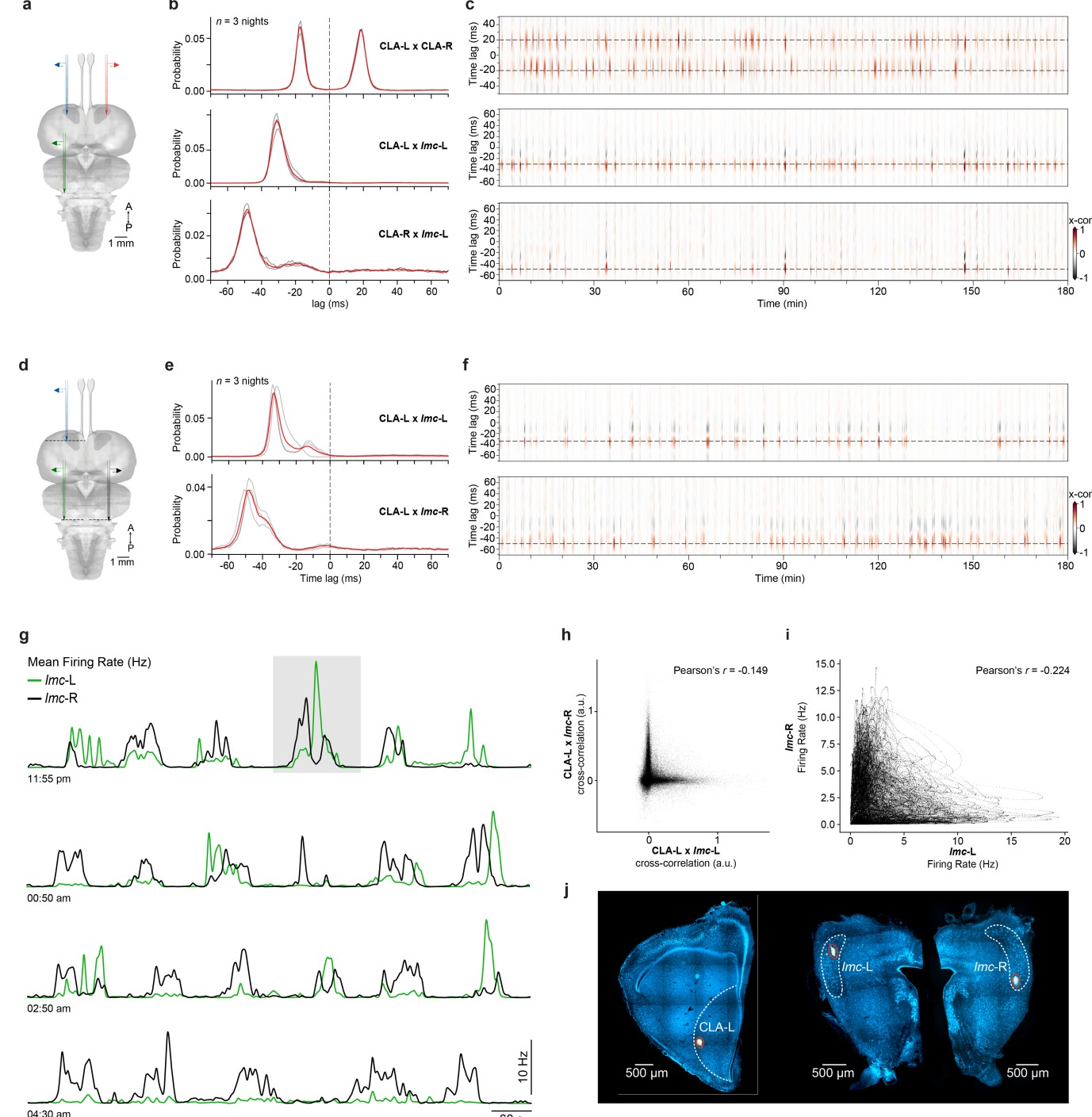

**Extended Data Fig. 7** | See next page for caption.

**Extended Data Fig. 7 | Imc recordings. a**, Schematic of the recording configuration, with two Neuropixels probes in claustrum bilaterally, and one probe inserted into the left Imc. **b**, Distributions of peak-correlation lags between left and right claustra (top), left claustrum and left Imc (middle), and right claustrum and left Imc (bottom). Data for three nights of the same animal; average plotted in red. **c**, Cross-correlations for one of the nights and 3 h of sleep corresponding to the histograms in **b** (rows). Note the bands of maximum correlation, highlighted with a stippled line and corresponding to the peaks of the histograms shown in **b**. **d**, Schematic of the recording configuration, showing one probe in left claustrum, and two further probes implanted into the Imc bilaterally. Stippled lines indicate approximate locations at which the sections shown in **j** have been taken from. **e**, Distributions of peak-correlation lags between left claustrum and left Imc (top), and left claustrum and right Imc. Data for three nights; average plotted in red. **f**, Cross-correlations for one night and three hours of sleep corresponding to the histograms in **e** (rows). **g**, Mean firing rates of putative Imc units, recorded bilaterally, and shown for different time periods during one night. Note the periodic increases in spiking activity that correspond to individual $REM_P$ cycles, separated by periods of relatively low activity in left (green) and right (black) Imc, corresponding to SW. The shaded area highlights the $REM_P$ cycle shown in Fig. 4k. Also note the persistently higher firing rate of the right Imc during five consecutive $REM_P$ cycles in the period starting at 4:30 a.m. (bottom panel). This is consistent with the observation that one claustrum tends to dominate and lead for multiple cycles in the last 2–3 h of the night, as shown in Fig. 3d. **h**, Values of the cross-correlation between left claustrum and ipsilateral versus contralateral Imc. Each dot represents a sample taken every 100 ms during $REM_P$ sleep ($n = 130,911$; 9 h of sleep). Note that samples are distributed along the axes, with an absence of points in the upper right quadrant (1,1), indicating mutual exclusion. Pearson's $r = -0.149$, $P = 0.0$. Compare also with **f**, in which CLA activity correlates with only one Imc and not both at the same time. **i**, Firing rates of left and right Imc. Each dot represents a sample taken every 100 ms during $REM_P$ sleep ($n = 130,911$; 9 h of sleep). Pearson's $r = -0.224$, $P = 0.0$. **j**, Fluorescent Nissl stains of transverse sections, highlighting recording sites in the left claustrum and in the Imc bilaterally.

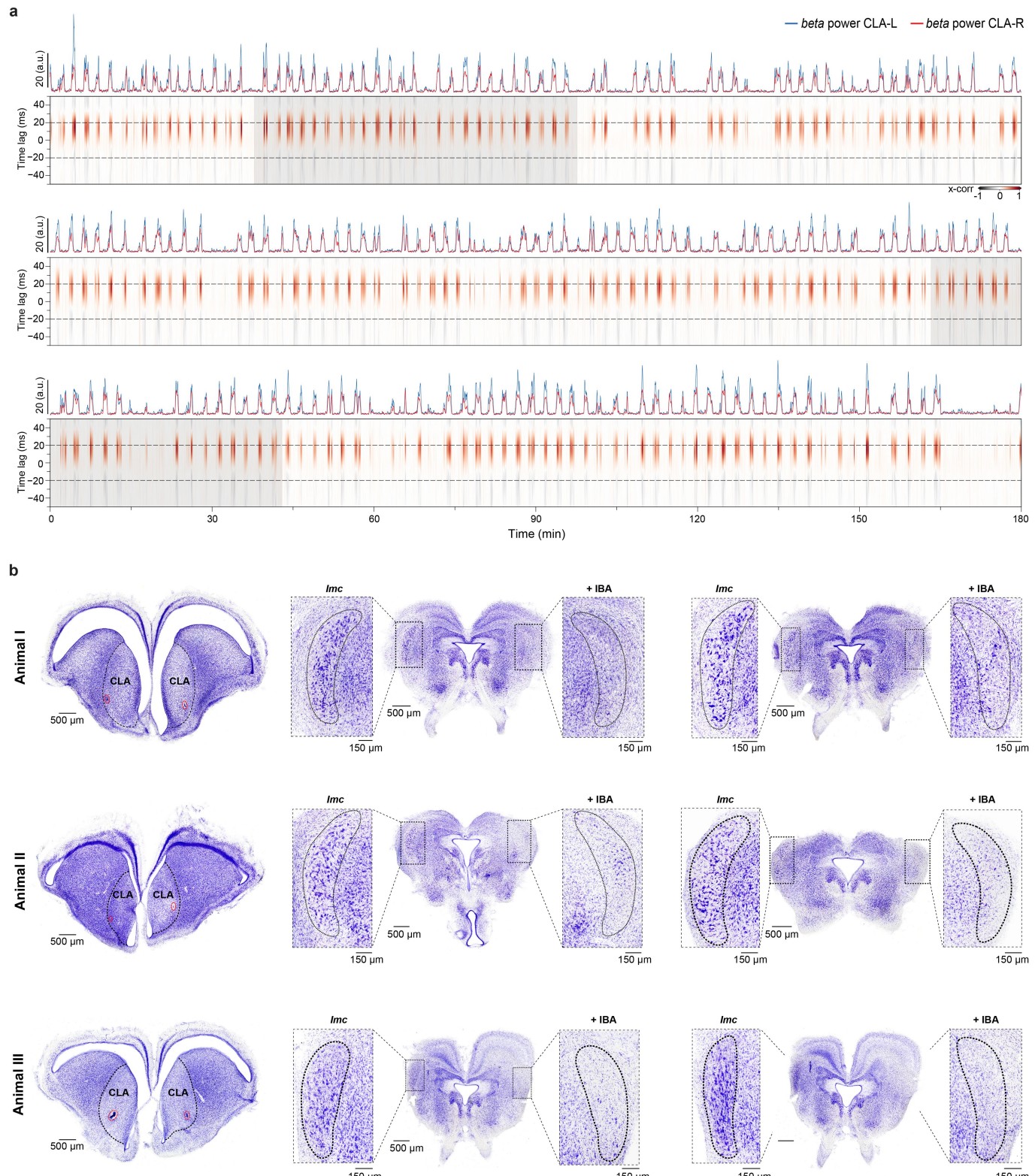

**Extended Data Fig. 8 | Further characterization of Imc lesions. a**, Unilateral Imc lesion. Shown are the cross-correlations between left and right claustra for 9 h of sleep, together with the normalized beta power for both claustra above. Shaded areas indicate the time periods shown in Fig. 5c. **b**, Nissl stains of transverse sections at the level of the claustrum (left column) and the Imc (middle and right column), for three animals with unilateral IBA-induced lesions. Note the Imc lesions and cell-body loss after injection of IBA, and the recording sites in the claustra, as identified by electrolytic lesions (red stippled line).

| | |
|---|---|

# Reporting Summary

## Statistics

For all statistical analyses, confirm that the following items are present in the figure legend, table legend, main text, or Methods section.

| n/a | Confirmed | |
|---|---|---|
| ☐ | ☒ | The exact sample size (*n*) for each experimental group/condition, given as a discrete number and unit of measurement |
| ☐ | ☒ | A statement on whether measurements were taken from distinct samples or whether the same sample was measured repeatedly |
| ☐ | ☒ | The statistical test(s) used AND whether they are one- or two-sided<br>*Only common tests should be described solely by name; describe more complex techniques in the Methods section.* |
| ☒ | ☐ | A description of all covariates tested |
| ☒ | ☐ | A description of any assumptions or corrections, such as tests of normality and adjustment for multiple comparisons |
| ☐ | ☒ | A full description of the statistical parameters including central tendency (e.g. means) or other basic estimates (e.g. regression coefficient) AND variation (e.g. standard deviation) or associated estimates of uncertainty (e.g. confidence intervals) |
| ☐ | ☒ | For null hypothesis testing, the test statistic (e.g. *F*, *t*, *r*) with confidence intervals, effect sizes, degrees of freedom and *P* value noted<br>*Give P values as exact values whenever suitable.* |
| ☒ | ☐ | For Bayesian analysis, information on the choice of priors and Markov chain Monte Carlo settings |
| ☒ | ☐ | For hierarchical and complex designs, identification of the appropriate level for tests and full reporting of outcomes |
| ☐ | ☒ | Estimates of effect sizes (e.g. Cohen's *d*, Pearson's *r*), indicating how they were calculated |

*Our web collection on statistics for biologists contains articles on many of the points above.*

## Software and code

Policy information about availability of computer code

| | |
|---|---|
| Data collection | SpikeGLX for Neuropixels recordings (http://billkarsh.github.io/SpikeGLX/);<br>Cheetah (Neuralynx) for recordings using 32-channel NeuroNexus probes.<br>Zen 2.1 and 3.1 (Carl Zeiss) was used for image acquisition. |
| Data analysis | Python (version 3.8), and MATLAB (MathWorks) version R2019b.<br>Python packages used: scipy (1.6.2), numpy (1.20.3), pandas (1.3.0), and xarray (0.18.2).<br><br>Neuropixels recordings: spike sorting was performed with Kilosort2 (https://github.com/MouseLand/Kilosort), using the ecephys_spike_sorting package (https://github.com/jenniferColonell/ecephys_spike_sorting); Phy was used for manual curation (https://github.com/cortex-lab/phy).<br><br>Recordings with NeuroNexus probes (32-channels): Ironclust (https://github.com/flatironinstitute/ironclust) was used for spike sorting and manual curation. |

For manuscripts utilizing custom algorithms or software that are central to the research but not yet described in published literature, software must be made available to editors and reviewers. We strongly encourage code deposition in a community repository (e.g. GitHub). See the Nature Portfolio guidelines for submitting code & software for further information.

## Data

Policy information about availability of data

All manuscripts must include a data availability statement. This statement should provide the following information, where applicable:

- Accession codes, unique identifiers, or web links for publicly available datasets
- A description of any restrictions on data availability
- For clinical datasets or third party data, please ensure that the statement adheres to our policy

> Data will be available upon reasonable request.

## Human research participants

Policy information about studies involving human research participants and Sex and Gender in Research.

| | |
|---|---|
| Reporting on sex and gender | *Use the terms sex (biological attribute) and gender (shaped by social and cultural circumstances) carefully in order to avoid confusing both terms. Indicate if findings apply to only one sex or gender; describe whether sex and gender were considered in study design whether sex and/or gender was determined based on self-reporting or assigned and methods used. Provide in the source data disaggregated sex and gender data where this information has been collected, and consent has been obtained for sharing of individual-level data; provide overall numbers in this Reporting Summary. Please state if this information has not been collected. Report sex- and gender-based analyses where performed, justify reasons for lack of sex- and gender-based analysis.* |
| Population characteristics | *Describe the covariate-relevant population characteristics of the human research participants (e.g. age, genotypic information, past and current diagnosis and treatment categories). If you filled out the behavioural & social sciences study design questions and have nothing to add here, write "See above."* |
| Recruitment | *Describe how participants were recruited. Outline any potential self-selection bias or other biases that may be present and how these are likely to impact results.* |
| Ethics oversight | *Identify the organization(s) that approved the study protocol.* |

Note that full information on the approval of the study protocol must also be provided in the manuscript.

# Field-specific reporting

Please select the one below that is the best fit for your research. If you are not sure, read the appropriate sections before making your selection.

☒ Life sciences ☐ Behavioural & social sciences ☐ Ecological, evolutionary & environmental sciences

For a reference copy of the document with all sections, see nature.com/documents/nr-reporting-summary-flat.pdf

# Life sciences study design

All studies must disclose on these points even when the disclosure is negative.

| | |
|---|---|
| Sample size | No statistical tests were used to predetermine sample sizes. We established that our sample sizes are sufficient based on previous experience and commonly used sample sizes in this field of research, taking into account the unusual nature and limited availability of the animal species studied. |
| Data exclusions | Experiments with off-target placement of electrodes, ibotenic acid or tracer injections were excluded from our analysis. |
| Replication | We could replicate all our results. |
| Randomization | Animals were not assigned to groups, and were selected based on weight and healthy appearance. Randomization was not relevant for our study. |
| Blinding | Investigators were not blinded to group allocation during data collection and analysis. Our study was mostly observational in nature with the exception of lesion experiments (Fig. 5). Measurements were fully automated, and blinding was thus not relevant for our study. |

# Reporting for specific materials, systems and methods

We require information from authors about some types of materials, experimental systems and methods used in many studies. Here, indicate whether each material, system or method listed is relevant to your study. If you are not sure if a list item applies to your research, read the appropriate section before selecting a response.

## Materials & experimental systems

| n/a | Involved in the study |
|---|---|
| ☐ | ☒ Antibodies |
| ☒ | ☐ Eukaryotic cell lines |
| ☒ | ☐ Palaeontology and archaeology |
| ☐ | ☒ Animals and other organisms |
| ☒ | ☐ Clinical data |
| ☒ | ☐ Dual use research of concern |

## Methods

| n/a | Involved in the study |
|---|---|
| ☒ | ☐ ChIP-seq |
| ☒ | ☐ Flow cytometry |
| ☒ | ☐ MRI-based neuroimaging |

## Antibodies

| Antibodies used | Primary antibodies: anti-ChAT (1:500, Invitrogen, MA5-31383);  anti-Hippocalcin (1:1000, abcam, ab24560)<br>Secondary antibodies: Donkey anti-mouse or anti-rabbit, conjugated with Alexa Fluor 488 (1:500, Invitrogen, A-21202 and A-21206) |
|---|---|
| Validation | anti-Hippocalcin: the manufacturer validated the antibody by Western blot (abcam).<br>anti-ChAT: Example images shown on product webpage (ThermoFisher): cholinergic neurons in the caudate putamen (IHC), ICC/IF analysis of ChAT in SK-MEL-30 cells, and IHC analysis of ChAT in the human plancenta (cells in chorionic villi).<br>IHC validation was performed in our laboratory, testing various concentrations on lizard tissue. |

## Animals and other research organisms

Policy information about underline{studies involving animals}; underline{ARRIVE guidelines} recommended for reporting animal research, and underline{Sex and Gender in Research}

| Laboratory animals | Adult Lizards (Pogona vitticeps), weighing 150-250g, bred and housed in our state-of-the-art animal facility. |
|---|---|
| Wild animals | This study didn't involve wild animals. |
| Reporting on sex | Our findings apply to either sex, which has been assigned based on visual and/or ultrasound inspection. Experimental animals have been chosen based on availability, weight and healthy appearance, irrespective of their sex. We did not perform any sex-based analysis, but our results are highly consistent across all animals studied. |
| Field-collected samples | This study did not involve field-collected samples. |
| Ethics oversight | All experimental procedures were approved by the relevant animal welfare authority (Regierungspräsidium Darmstadt, Germany) and conducted following the strict federal guidelines for the use and care of laboratory animals (permit numbers V54-19c20/15-F126/1005_1011 and 2006). |

Note that full information on the approval of the study protocol must also be provided in the manuscript.

