## [Peer Review File · Nature]

Manuscript Title: Inter-hemispheric competition during sleep

Reviewer Comments & Author Rebuttals

Reviewer Reports on the Initial Version:

Referees' comments:

Referee #1 (Remarks to the Author):

This manuscript by Fenk et al show that the isthmus complex of Australian sleeping dragon controls the coordination of REM local field potentials in the claustrum. Previously, the authors showed that lesioning the claustrum specifically reduced SWS sharp waves but not REM sleep patterns recorded in the anterior dorsal ventricular ridge. The experiments in the current manuscript are excellent, and data are high quality. Although the experiments themselves describe an interesting observation, the manuscript does not provide a complete picture of what this circuit does. Does this connection have a role in REM-dependent memory consolidation? The lmc comprises a set of nuclei with different functions in birds and mammals, so are there specific lmc subregions in the reptile and different cell types that are mediating this connection? Does the lmc have a general role in coordinating hemispheres beyond the reptile claustrum? What function does this interhemispheric delay serve? What is the circuit mechanism mediating the communication from the lmc to the claustrum? In its current form, the manuscript is too descriptive and preliminary and would need to be expanded on with an enormous number of additional experiments to be suitable for this forum.

The authors state that the reptile may be a useful model system, which I agree with, but I'm not sure that statement is warranted here with respect to the circuits under investigation. While there is no doubt that studying sleep in reptiles may yield some useful evolutionary insight into sleep function, I am not convinced the neural architecture can be compared and results from reptiles extrapolated here. The claustrum in mammals is not well understood (particularly during REM), so making an evolutionary comparison here seems uninformed. A similar concern arises with the isthmus complex which has been studied very well in the avian brain in the context of visual attention. However, it has not been well investigated in sleep in birds, and in mammals the isthmus complex is recognized as a set of brain regions and subnuclei each with a highly divergent function.

With respect to this previous point on lmc, the mammalian equivalent of the isthmus and lmc is a diverse set of nuclei including tegmental nucleus, parts of raphe nucleus, substantia nigra, and even the periaqueductal gray (each of which have been investigated independently as the authors know). Therefore, it seems this is a very heterogeneous set of regions which would require a more thorough description at the molecular and anatomical scale, how this cellular heterogeneity relates to connectivity. There is mention that cholinergic, glutamatergic, and GABAergic cells reside in the lmc, and then the discussion treats the lmc as an inhibitory nucleus. However, this is at odds with a 20ms delay from lmc activity to claustrum activity which would presumably be supported an excitatory connection (although no physiology was performed in this connection). In conclusion, the work on this pathway is far too preliminary as presented.

The authors previously showed that the claustrum is not involved in the generation of REM sleep patterns, so it is unclear what this lmc claustrum connection is doing functionally.

The inconsistent and sparse use of statistics in this manuscript makes it frustrating to evaluate. It reads wonderfully, but evaluation is different. For example, in Figure 1c there is an n=190,578 waves with no mention of session number, animal number, etc. Sometimes the mean value is shown without a measure of variance, in other instances, p values only. I could comment on every aspect of the results and figure panels here, but it would unnecessarily belabor this point. The paper is descriptive, but the lack of rigorous description does not instill a sense of confidence when trying to evaluate the manuscript.

The authors mentioned that the rationale for the work on the lmc was from tracing. What tracing are they referring to? I do not see any in this manuscript.

As in almost all sleep related papers, it would be good to have time-frequency (Fourier analysis) analyses accompanying all figures and included in analyses to observe the dynamics as a function of time. The raw data traces shown do not provide any meaningful information beyond the global ~ 100s periods of alternating LFP amplitude.

Referee #2 (Remarks to the Author):

This is an elegant study that uncovers new phenomena and mechanisms of sleep in the lizard *Pogona*. The major focus of the work is REM sleep, and the interhemispheric coordination between the brain activity during REM sleep. Using simultaneous multi-site recordings with multiple silicon probes, the claustrum and midbrain/hindbrain area lmc are shown to have coordinated activity during REM. In particular, one hemisphere's claustrum is shown to lead/dominate the other, the leading side switches from time to time at a slow timescale, the lmc is shown to have alternating activity between hemispheres, and lesions show that which claustrum leads/dominates depends on the lmc. Both these structures (claustrum and lmc) have homologous areas in mammals as well as other species, and therefore the findings from this simpler brain have implications for understanding sleep in mammals and other species.

This work adds to the recent findings linking the claustrum to sleep-related processes in mammals and lizards. Previously, this linkage was made for NREM-related activity (done by the same group in the lizard), and here is done for REM-related activity. (This study also adds phenomenological findings of hemispheric activity during NREM in claustrum.) Because the claustrum is extensively connected to many parts of the brain, claustrum activity should significantly impact brain-wide activity during REM sleep too. Related to this, it would be very interesting to see what other parts of the brain are doing during REM sleep, specifically how the activity may relate to the strong, irregular SN events in claustrum – only if the authors happen to have simultaneous recordings from other areas during the recordings that they already made from the claustrum.

One other comment relates to the topic of competition between hemispheres. It's not clear to me that the alternating leading/lagging activity in the claustrum during REM sleep suggests a competitive process per se between the claustrum of each hemisphere. This is because the lagging claustrum's activity closely matches the leading one's with a short lag, and that does not necessarily imply competition, at least in the way that I usually think of neural competition (this is a minor point, but something that made we wonder what kind of competitive process was suggested while reading the first part of the paper before the lmc experiments). The clearest evidence of competition was the alternating activity between the two lmc's. Related to this, I think the scatterplot in Ext. Data Fig 6i might be clearer if only the periods of REM sleep were included, then perhaps a negative correlation would be more visible. It may also help to normalize the firing rates during each REM episode before compiling them across episodes.

Overall, this is very nice work on a new model system for studying fundamental sleep-related processes during the main sleep states of REM and NREM sleep.

Referee #3 (Remarks to the Author):

In this article Fenk et al. studied the neuronal networks and electrophysiological mechanisms of sleep in the bearded dragon lizard. They notably describe a new type of wave, the Sharp Negative (SN) waves, occurring during a mammalian REM sleep like state recorded in the equivalent of the claustrum. The authors nicely demonstrate, that in contrast to the sharp waves of the NREM sleep like state, SN waves occur in each hemisphere with a stable 20ms delay between the left and the right hemisphere. Each hemisphere leading alternatively under the control of a GABAergic midbrain nucleus (the lmc) suggesting a competition between the two hemispheres to generate these waves.

This new neuronal mechanism clearly suggests that functional difference exist between the two sleep states of the lizard. This work is highly relevant in a comparative context to understand which mechanism and function are species specific and how sleep evolved. Indeed, by describing in detail this phenomenon, its associated neuronal network and this unexpected interhemispheric competition occurring during the possible REM sleep like state, Fenk et al. provide a new basis to draw parallels and reveal difference between species, which is crucial to understand the species specific and core functions of sleep. Many sleep-related brain oscillations have been implicated in processing memories in mammals, but little is known about the role brain oscillations play in this process in non-mammals. Consequently, this work on dragons opens a new opportunity to examine this question from a comparative perspective. In addition, this discovery would be of immediate interest to the field as the comparative approach, as there is a trend in the field to move away our mammalian centric view of sleep.

The authors use up-to-date methods for recording and analyzing the data. The figures are extremely clear and illustrate nicely all the different steps at different timescales from individual to average data. I also found that the experiment is well conceived demonstrating the existence of a new type of brain waves during sleep in lizard (SN waves), their generation and modulation. The interhemispheric competition described in the paper is highly stereotypical and convincing. The

supplementary material showing individual data and its repeatability across individuals is very useful. The immunostaining and brain lesioning look clear and specific. In conclusion, I find the methods detailed enough to replicate the data.

My main concern about the paper is not related to the quality and reliability of the methods and results, but rather on the terminology and the broad interspecies interpretation. Indeed, whereas the mechanistic interpretation is clear and strong, the authors use mammal-based terminology (NREM and REM sleep) when describing the sleep states and electrophysiological features of sleep in dragons. Despite the fact that this group already published articles on sleep in lizards with such terminology, it is still not clear at what point the two states found in the bearded dragon are homologous / analogous or similar to the mammalian sleep states, as large differences remain between the two groups (See Blumberg et al 2020, Libourel & Barrillot 2020). This is particularly important for the specific terminology of the sleep waves, like sleep spindles, Sharp Wave ripples complex, delta waves or slow waves. All these waves refer to specific neuronal networks, associated with specific function, like memory consolidation, and none of them strictly fit with the waves found in reptiles in term of shape, localization and frequency. Using mammalian terminology in non-mammalian species based on similar morphology alone is clearly not enough to infer the homology the use of the same term implies. and the use of mammalian terms clearly biases the cross-species interpretation of the data, and undercuts the true power of comparative research.

To avoid the problems associated with using mammal-based names in non-mammalian species, the authors should find a way to revise the terminology (NREM, REM SWr). The author could refer to "REM sleep" as "REM sleep like state" or "NREM sleep" to "reptilian NREM sleep", and "SWr" as "SWr-like" or "reptilian SWr", in the whole text, abstract and title. I actually noticed that the author did this for the waves found during the REM sleep like state by naming them SN. As an example, if these waves would have been named Slow oscillations (SO) because they look like mammalian SO, this would have completely changed our interpretation of the lizard REM sleep like state and possibly our interpretation of the evolution of sleep. By no means does this criticism of the terminology undercut the importance of the data presented.

I have also several other comments on the papers listed below:

- From line 49 to 60 the introduction on sleep in reptiles should be fully revised accordingly to my main comment regarding terminology.
- Line 67: the authors specify that the recordings were made from the claustrum equivalent or the adjacent anterior DVR. Are there any differences between the recording sites in terms of SN, spikes, delay?
- How much does the LFP signal vary as a function of depth? A representation of the LFP traces from the different electrode sites and brain regions would be very useful. This could illustrate if there is any phase reversal recorded from the silicon probe showing a local phenomenon like that shown for the reptilian SWr (Shein Idelson et al 2016). The author should provide more information about the regional aspect of the SN.
- It would be very informative to provide more information regarding the SN extracellular unit discharge. Over all the animals and electrode sites, how many SN were phase locked with unit activity. It is actually not clear me to how reliably the neuronal discharge is linked with the SN. Is the figure 1 panel e showing only one electrode site from one animal? This should be clarified.

- One of the major findings of this paper is the inter-hemispherical coordination of the SN during REM sleep like state, in opposition to the SWr of NREM sleep like state which are not synchronize across hemisphere. This is an important feature to assume that the SN and SWr are generated by different neuronal processes, possibly underlying different function. However, only figure 2 panel b illustrated that purpose. But how many animals, nights, samples do this correspond to? We have actually no idea about the reliability of the absence of coordination. It would help to provide more information on this.

- The lmc lesion experiment is highly convincing. I wonder if this lesion also influences the reptilian SWr occurrence, amplitude?

As the authors state in the discussion, I also wonder whether this lmc claustra-coordination exists during wake. As the authors recorded for hours even after lights on, I wonder whether they have enough wake to provide this information? This would be indeed of great interest to see if differences exist between wake and the REM sleep like state.

- The author should define WTA and STDP, as the abbreviations are used without definition.

- At what temperature were the animals recorded?

- In the SN detection and matching methods, how was noise estimated (line 423).

- Line 403: how were the channels of the probes chosen?

- Fig 4c (line 741) where are the "corresponding levels"? This is not clear.

- The following references are missing but highly relevant to the paper:

1- Wang et al 2020, "Subdivisions of the mesencephalon and isthmus in the lizard *Gekko gecko* as revealed by ChAT immunohistochemistry",

2- Ghosh et al 2022, "Running speed and REM sleep control two distinct modes of rapid interhemispheric communication"

In particular Ghosh et al 2022 should be discussed regarding the finding of the paper.

Paul-Antoine Libourel.

Author Rebuttals to Initial Comments:

Revisions of Nature manuscript 2022-06-10161 (Fenk et al.)

We thank the reviewers for their constructive comments on our manuscript. We address them individually below.

Referee #1 (Remarks to the Author):

This manuscript by Fenk et al. show that the isthmus complex of Australian sleeping dragon controls the coordination of REM local field potentials in the claustrum. Previously, the authors showed that lesioning the claustrum specifically reduced SWS sharp waves but not REM sleep patterns recorded in the anterior dorsal ventricular ridge. The experiments in the current manuscript are excellent, and data are high quality.

We thank the reviewer for this appreciation.

1. Although the experiments themselves describe an interesting observation, the manuscript does not provide a complete picture of what this circuit does.

We think that it is rare, assuming it is ever true, that a paper provides a complete picture of what a circuit does. We will lay out in this revision the multiple ways in which what we present is novel, surprising and significant, revealing new dynamics, new connections and new actions of central but ill-understood vertebrate brain structures, during a specific phase of sleep. The other reviewers seem to agree with us concerning the novelty and importance of our results.

2. Does this connection have a role in REM-dependent memory consolidation?

This is an interesting question, and one which we asked and still ask in our discussion section (lines 294-297 of original submission). We do not yet know the answer to it, but are indeed interested in it; this will be the object of further studies, requiring a completely different set of experiments.

3. The lmc comprises a set of nuclei with different functions in birds and mammals, so are there specific lmc subregions in the reptile and different cell types that are mediating this connection?

*We do not fully understand the question. The Isthmic complex in *Pogona* is composed of several nuclei, each with specific neurotransmitters and connections; the lmc is one of them, as we described in detail in our original manuscript (in Extended Data Figure 5, now Extended Data Fig. 6a-g, see also Fig. 4b-d), and reproduced again below. This organization appears to be very similar, if not identical, to that described in birds before (e.g.¹⁻⁵).*

New Extended Data Fig. 6. The isthmic nuclei (Imc and lpc) and their connections. **a.** Nissl stain of a horizontal section of the *Pogona* brain. Imc and lpc are located at the mid- to hindbrain junction, and are highlighted on the right hemisphere (black stippled ellipsoids). **b.** Fluorescent Nissl stain (blue) of a horizontal section through the mesencephalon, plus the lpc (midbrain-hindbrain transition). Note the small, cholinergic cell bodies characteristic of the lpc (green), and cholinergic fibers extending into the optic tectum (OT, superior colliculus in mammals). A neurobiotin injection labeled large cell bodies in the Imc (red), whose axons innervate the lpc. **c.** Retrograde labeling of cell bodies in the Imc (left) and lpc (right) following injection of Neurobiotin into the OT. **d.** Retrograde labeling of cell bodies in the Imc (left) and OT (right) following injection of Neurobiotin into the lpc. **e.** Retrograde labeling of cell bodies in the OT (left) and lpc (right) following injection of Neurobiotin into the Imc. **f.** Schematic of the brainstem with Vglut2 (yellow) and Vgat (blue) staining. **g.** Schematic diagram of the connections between OT, Imc, and lpc, with labels for V, III, V, VI, I, IV, Vglut, Pvalb, ChAT, and Vglut.

following injection of Neurobiotin into the *lpc*. Also note the fine ramifications of neural processes in the OT, likely corresponding to “paintbrush” axon terminals of the *lpc* described by Cajal (1911), and later shown to provide focal, re-entrant amplification of retinal input to the OT. **e.** Retrograde labeling of cell bodies in the OT, following injection of Neurobiotin into the *lmc*. **f.** Sagittal section through the lateral mesencephalon. Double in situ hybridization with *Vglut2* and *Vgat* probes. **g.** Summary diagram of the ipsilateral connections shown in **b-e**; expression of marker genes shown in **f** and Fig. 4c.

4. Does the *lmc* have a general role in coordinating hemispheres beyond the reptile claustrum?

As we cited and cite in the manuscript (see below: **), the principal and strongest evidence for *lmc* involvement in competitive interactions that we are aware of comes from work in birds, indicating a role in unilateral, bottom up, competition between inputs (mostly visual) in awake or anesthetized animals (sleep does not seem to have been studied), converging to the optic tectum^{3,5}. Unpublished work from the Mysore lab at Hopkins suggests that this type of bottom up competition can be between the two eyes but we are not aware of published work demonstrating the role of *lmc* in bilateral competition.

[**: “*lmc* is part of a complex of cholinergic/glutamatergic and GABAergic isthmic nuclei best studied in birds^{31,33,35,39} and implicated in bottom-up attention: *lmc*, embryologically a pre-isthmic nucleus⁴⁰, mediates a type of broad lateral inhibition of excitatory neurons in both the ipsilateral optic tectum and *lpc*, a companion cholinergic/glutamatergic nucleus that forms a positive feedback loop with the ipsilateral tectum^{39,41}. When two stimuli fall onto one retina, these circuits undergo a WTA competitive interaction such that responses to the stronger stimulus are selected, and attention and gaze are directed towards it, rather than to a weighted mean of the two stimuli. *lmc*, by virtue of its heterotypic connectivity with OT and *lpc*, underlies this competition^{32,33}. This work initially concerned the selection of competing visual stimuli falling on the same retina, but recent experiments established competition also between contralateral stimuli, and even between sensory modalities⁴², although the underlying circuits are not well understood. Homologues of these avian (reptilian) isthmic nuclei are found in fish^{43,44}, amphibians^{45,46}, non-avian reptiles^{47,48} and in mammals (*parabigeminal* nucleus), where a role in visual attention has been hypothesized⁴⁹.”]

Recent work in zebrafish (Fernandes et al., *Neuron*, 2021), points to an inhibitory nucleus isthmi playing a related role in visual stimulus selection. We now cite this paper also. (<https://www.sciencedirect.com/science/article/pii/S0896627320309612>)

Collicular competition in rodents has also been discussed recently by Lee and Sabatini (*Nature* 2021, <https://www.nature.com/articles/s41586-021-04055-4>), but in a different context.

5. What function does this interhemispheric delay serve?

This is, as in 2 above, the reframing of a question that we asked in our discussion section (line 290 in original ms.): “Third, what is the functional significance of the tight correlation and ± 20 ms delays between claustra during REM?”. In the discussion, we proposed a hypothesis (that it might serve as a substrate for STDP) that has yet to be tested, as in our answer to point 2 above. This hypothesis is not the object of our study, but a follow up question which we or others will need to test.

We note that the question of functionality is a complex and delicate one. We prefer to treat the delay as an objective, factual observation (the earlier side is also the one that dominates over the other, and thus the delay represents, to an observer, which of the two sides of the brain dominates the other). Whether the delay (probably due to conduction and synaptic transfer) plays a “function” (understood as: “has functional consequences”; e.g., as a putative substrate for STDP), is a new question (which

we pose in the discussion), requiring new experiments and tests. We are indeed very interested in the answers, but consider that this question represents a new project derived from our results, as are another six questions that we asked.

6. What is the circuit mechanism mediating the communication from the lmc to the claustrum?

This again reframes two of our own questions: (discussion, lines 285, 287 of original ms.) "First, what pathways underlie the bilateral mesencephalic competition (Fig. 5e)... Second, what pathways link lmc, a GABAergic nucleus, to the activation of both claustra?", questions which we evidently identify as direct extensions of our results.

We provide direct functional evidence that a competitive phenomenon observed between the two claustra in the forebrain during REM depends on the integrity of the lmc in the midbrain tegmentum. The details of the cellular and synaptic circuit components linking them are obviously interesting, but they are of a different nature, and are not necessary to appreciate the results and the significance of our study. Said differently, knowing what those components are is a natural extension of our results, but would not, absent our results, lead to the prediction that L and R claustra compete with one another during sleep and even less during REM (the main result of our work).

7. In its current form, the manuscript is too descriptive and preliminary and would need to be expanded on with an enormous number of additional experiments to be suitable for this forum.

We respectfully disagree. First, being descriptive is the first requirement for establishing experimental facts. Second, our paper goes well beyond being simply descriptive and preliminary, for it establishes, via specific manipulations, a necessary functional link between two parts of the brains (the claustrum and the midbrain tegmental isthmus), not known previously to interact with each other, in a bilateral competition between brain hemispheres during sleep (not previously known), and only during one phase of sleep (also not previously known). Our results establish that the integrity of the lsm is required for this bilateral competition to take place. Our results also rule out any role for forebrain structures in the bilateral competition between claustra. (Our "descriptive and preliminary results" do result from a very large number of functional and anatomical experiments.)

8. The authors state that the reptile may be a useful model system, which I agree with, but I'm not sure that statement is warranted here with respect to the circuits under investigation. While there is no doubt that studying sleep in reptiles may yield some useful evolutionary insight into sleep function, I am not convinced the neural architecture can be compared and results from reptiles extrapolated here. The claustrum in mammals is not well understood (particularly during REM), so making an evolutionary comparison here seems uninformed.

We do not understand the reviewer's comment and objections, which are also at odds with those of the other reviewers. We state in our introduction: "If so, comparative approaches in systems that represent diverse animal lineages might help better understand not only the evolution of sleep but also its functions and mechanistic underpinnings." We then present our results which are, to our knowledge, both novel and unexpected in any system.

We do not state anywhere that our results apply to the mammalian brain; we believe, however, that our results could lead to targeted investigations testing these results in mammals as well. Our recent work on lizard claustrum and NREM sleep (Norimoto, Fenk et al., 2020) converged with mammalian results published shortly after our own (Narikiyo et al., 2020), indicating that the comparative approach is indeed both useful and instructive.

Note also Reviewers #2 and #3's comments, supporting our strategies:

For example, R#2 writes: "Both these structures (claustrum and lmc) have homologous areas in mammals as well as other species, and therefore the findings from this simpler brain have implications for understanding sleep in mammals and other species."

And R#3 write: "...Fenk et al. provide a new basis to draw parallels and reveal differences between species, which is crucial to understand the species specific and core functions of sleep. Many sleep-related brain oscillations have been implicated in processing memories in mammals, but little is known about the role brain oscillations play in this process in non-mammals. Consequently, this work on dragons opens a new opportunity to examine this question from a comparative perspective. In addition, this discovery would be of immediate interest to the field as the comparative approach, as there is a trend in the field to move away our mammalian centric view of sleep."

The advantages of comparative approaches are many. Among them is the possibility to identify similarities and differences between systems, and link those to structural (or other) similarities and differences. These comparisons provide a useful tool to establish or confirm causality, and to identify shared or divergent functional principles.

Finally, the fact that "the mammalian claustrum is not well understood (particularly during REM)", as the reviewer states, should not be a valid argument to avoid studying the claustrum in other species. We think, on the contrary, that it is a good argument to examine claustral activity and function in species in which this can, for example, be done more easily—as is the case with *Pogona* in which the claustrum is not a thin sheet, difficult to target precisely and record from.

9. A similar concern arises with the isthmic complex which has been studied very well in the avian brain in the context of visual attention. However, it has not been well investigated in sleep in birds, and in mammals the isthmic complex is recognized as a set of brain regions and subnuclei each with a highly divergent function.

Here also, the reviewer seems to object to the fact that our results have not been established in other systems (avian or mammalian) already. This criticism seems unfair. If our results had been reported in other systems already, they would lose their novelty.

Similarly, we are not aware of published mammalian work examining isthmus activity during sleep, or its role in controlling L-R interactions in the forebrain. We thus believe that imputing "highly divergent functions" is not warranted by existing evidence. An advantage of broad systems approaches is precisely that they sometimes uncover new phenomena or functions not examined broadly.

Finally, biological results in non mammalian species do not necessarily get their value from being applicable to mammals. Similarly, biological results in certain mammalian species (e.g., rodents) do not necessarily get their value from being applicable to other mammalian species (e.g., primates, including humans). We think that it is through such comparisons that one can establish the existence of shared principles, identify the sites of action of evolutionary variations, and understand the relationships between circuit structure and circuit functions.

10. With respect to this previous point on lmc, the mammalian equivalent of the isthmus and lmc is a diverse set of nuclei including tegmental nucleus, parts of raphe nucleus, substantia nigra, and even the periaqueductal gray (each of which have been investigated independently as the authors know). Therefore, it seems this is a very heterogenous set of regions which would require a more thorough description at the molecular and anatomical scale, how this cellular heterogeneity relates to

connectivity. There is mention that cholinergic, glutamatergic, and GABAergic cells reside in the lmc, and then the discussion treats the lmc as an inhibitory nucleus.

The reviewer is mistaken. We do not mention that cholinergic, glutamatergic and GABAergic cells reside in the lmc. We wrote: "lmc is part of a complex of cholinergic/glutamatergic and GABAergic isthmic nuclei best studied in birds". Which is correct. The anatomy of the Isthmic complex of Pogona was described in our manuscript (Extended Data Fig. 5, now Extended Data Fig. 6a-g and Fig. 4b-d). We included the evidence again above (response to reviewer point 3), with the relevant figures and details, hoping that this will answer the reviewer's questions. In our results, we indeed and correctly state that lmc is the magnocellular nucleus, characterized by its large GABAergic PV+ cells, as described also in avians^{2,4}. The lpc, or parvocellular nucleus, by contrast, contains cholinergic and glutamatergic nuclei, as also described in avians^{6,7}. In conclusion, lmc is indeed a GABAergic nucleus and our statement is correct.

11. However, this is at odds with a 20ms delay from lmc activity to claustrum activity which would presumably be supported an excitatory connection (although no physiology was performed in this connection). In conclusion, the work on this pathway is far too preliminary as presented.

It is indeed intriguing, and the object of two of the questions posed in our discussion (lines 285 to 290 of original ms). Answering them is not required, however, to prove the necessity of lmc's integrity for the competitive phenomenon that we demonstrate. This and similar requests (as in 6 above) point to interesting mechanistic questions that warrant further study; but answering them, as in response to point 6, would not lead to the functional predictions about sleep, about REM, about claustrum, about delays, about competition, that constitute our study (all novel).

12. The authors previously showed that the claustrum is not involved in the generation of REM sleep patterns, so it is unclear what this lmc claustrum connection is doing functionally.

We do not fully understand the reviewer's comment.

We reported in our earlier study (Norimoto, Fenk et al., 2020) that lesioning the claustrum suppressed sharp wave ripples observed in the forebrain (in regions downstream from the claustrum) during NREM but neither suppressed REM activity observed in those forebrain areas nor altered the REM/NREM alternation observed during sleep. This indicated that the generation of the alternating REM and NREM states does not depend on claustrum integrity.

We report here that activity recorded from the claustrum during REM originates upstream, that it is coordinated (with delays and competition) between L and R sides, and that this coordination is orchestrated in the midbrain by—at least in part, since its integrity is necessary—lmc. Indeed, lmc activity is remarkably correlated with—and may drive—that of the claustrum during REM (with a 30ms delay ipsilaterally, and 50ms contralaterally).

These two sets of results are consistent with each other.

13. The inconsistent and sparse use of statistics in this manuscript makes it frustrating to evaluate. It reads wonderfully, but evaluation is different. For example, in Figure 1c there is an n=190,578 waves with no mention of session number, animal number, etc. Sometimes the mean value is shown without a measure of variance, in other instances, p values only. I could comment on every aspect of the results and figure panels here, but it would unnecessarily belabor this point. The paper is descriptive, but the lack of rigorous description does not instill a sense of confidence when trying to evaluate the manuscript.

We respectfully disagree with this description and examine the reviewer's example chosen for criticism (Fig 1c) as illustration.

Concerning Fig 1c, we stated (main text): "Inter-event intervals (IEI), time-to-trough and amplitude distributions over about 190,000 events are shown in Fig 1c for one animal (from 9h of one night). The IEI distribution was skewed with a mode at about 40 ms (median: 60.2 ms; [25th, 75th] percentiles: [39.8, 110.5] ms)."

In addition, we provided the same measurements for 9 other recordings in Extended Data Figure 1a, which we provide again below.

We could not point to the mean values "without a measure of variance" that the reviewer refers to: whenever we provided a mean, it was in combination with examples to illustrate the variability of the data (e.g., Fig. 1b, Fig. 2b-d, Fig. 4g, Fig. 5d, Extended Data Figs 4a and c, 6b and e—now EDF5c,d and 7b,e). In addition, distributions are provided throughout (e.g., Fig 1c, 2f, 3b, 3e, 3f, EDF 1), with n values typically in the hundreds of thousands.

Concerning "p values only", we suspect that the reviewer is referring to the "p" we used as an axis-label in our plots showing the distributions of peak-correlation lags (Fig. 2d, Fig. 4i, Fig. 5d, Extended Data Figs. 4a and b, 6b and e—now EDF5c,d and 7b,e). It is indeed a probability, but not in the statistical sense of p-values that the reviewer mentions above. We now use the word "Probability" in that plot, to avoid confusion.

Note that the distributions of our data (see above for example) are not always Gaussian, hence the statistics we use.

The reviewer wishes that all of our statistics be formatted and provided in the same way, and we have done so where warranted in this revision. We took care, at the same time, to preserve readability, which we, like the reviewer, think is important.

We have added new statistical tests for the following statements and results:

- A Mann–Whitney U test on the difference of amplitude in relation to difference of lag for bilateral pairs of SN (new Fig 2f).
- A Wilcoxon signed-rank test on the different presence of positive and negative lags in our lesion studies (new Fig 5d).
- Updated Pearson’s coefficient of correlation for the mutual exclusion of bilateral Imc activity during REM periods (new Extended Data Fig. 7h and new Extended Data Fig. 7i).

14. The authors mentioned that the rationale for the work on the Imc was from tracing. What tracing are they referring to? I do not see any in this manuscript.

This is a misreading of our results and statements. Our tracings were not the rationale for the work on the Imc, and there is no evidence for direct Imc–CLA projections. What we wrote is that our retrograde tracings from the claustrum (see also Norimoto, Fenk et al., 2020) identified the amygdala and cortex as areas that could potentially underlie the L-R coordination. Lesioning the amygdala and cortex (together with parts of the DVR and striatum) did not prevent the bilateral switching of REM claustrum activities (new Extended Data Fig. 5). It was because of these negative results that we started to explore areas at the mid-to-hindbrain junction, given their known involvement in REM control in mammals. Our work on the Imc started with a serendipitous observation made while lowering our electrode through the midbrain and simultaneously monitoring claustrum activity.

It is correct that the claustrum retrograde tracings were not included in our original manuscript; we did not include them because what they suggested proved to be irrelevant for the inter-claustral phenomenon we describe. We have now nevertheless added some of these tracing results in Extended Data Fig. 5a (and below), at the reviewer’s request.

The Imc’s identity was established in part through tracing of connectivity (Extended Data Fig. 6).

Extended Data Fig. 5. a,b Identification of contralateral inputs and distribution of GFP-labeled neurons in the forebrain after injection of rAAV2-retro into the *Pogona* claustrum on one side. **a**, Transverse sections through the telencephalon, showing very sparse labeling in contralateral cortex (top), and dense labeling of neurons in the contralateral amygdala. **b**, Horizontal section through the tel- and mesencephalon, showing contralateral input from the amygdala. Also note the absence of labeled cells in the contralateral claustrum (within stippled line; pink fluorescence: hippocalcin)

15. As in almost all sleep related papers, it would be good to have time-frequency (Fourier analysis) analyses accompanying all figures and included in analyses to observe the dynamics as a function of time. The raw data traces shown do not provide any meaningful information beyond the global ~ 100s periods of alternating LFP amplitude.

We did not fully understand the reviewer's request. For example, what aspects and frequencies would he/she wish to see that might relate to the point of this manuscript? Indeed, the Fourier spectrum of these signals contains energy at frequencies from 1/ms (action potentials) to 1/hr (e.g., slow modulations in Fig 3d), i.e. over a range of at least 6 log units.

The stability of the *Pogona* sleep rhythm has been studied and reported in an earlier publication (Shein-Idelson et al., 2016) in its Figures 1, S3, S4, S5, S6, S7. We include Figure S3 of that paper below. This plot expresses sleep "dynamics as a function of time", expressed as delta over beta power ratios, throughout 6 nights of sleep and two animals. More examples are provided in the above reference.

Legend of Figure S3 in Shein-Idelson et al., 2016): Reliability of oscillation of high- δ and low- δ power epochs (defining E-sleep) across nights and animals. Electrophysiological sleep recorded from two lizards (#4 and 5) over three successive nights each. As in Fig 2E, each matrix plots δ/β power ratio (measured piecewise over 10s-long data segments, 1s steps) around night-time (time of night indicated at left). Rows along x represent 30-min segments running continuously (L to R) from top to bottom. Note that for all nights this high/low $[\delta/\beta]$ alternation starts shortly after light off (19:00hr), and continues uninterrupted until less than an hour before the end of behavioral sleep (although with smaller amplitude modulation in the last 2hrs, see Fig S4).

We added a spectrogram (in the band 0.1-100Hz) to accompany Fig. 2a in Extended Data Fig. 2a (and shown below). It shows the LFP data from the paired recording in Fig. 2a, with their band spectrograms. In it we also indicate the beta band, which we used to capture REM periods (see Methods).

The spectrograms adding no new information, we include them only in extended Fig 2a.

Referee #2 (Remarks to the Author):

This is an elegant study that uncovers new phenomena and mechanisms of sleep in the lizard *Pogona*. The major focus of the work is REM sleep, and the interhemispheric coordination between the brain activity during REM sleep. Using simultaneous multi-site recordings with multiple silicon probes, the claustrum and midbrain/hindbrain area *Imc* are shown to have coordinated activity during REM. In particular, one hemisphere's claustrum is shown to lead/dominate the other, the leading side switches from time to time at a slow timescale, the *Imc* is shown to have alternating activity between hemispheres, and lesions show that which claustrum leads/dominates depends on the *Imc*. Both these structures (claustrum and *Imc*) have homologous areas in mammals as well as other species, and therefore the findings from this simpler brain have implications for understanding sleep in mammals and other species.

We thank the reviewer for his/her comments.

This work adds to the recent findings linking the claustrum to sleep-related processes in mammals and lizards. Previously, this linkage was made for NREM-related activity (done by the same group in the lizard), and here is done for REM-related activity. (This study also adds phenomenological findings of hemispheric activity during NREM in claustrum.) Because the claustrum is extensively connected to many parts of the brain, claustrum activity should significantly impact brain-wide activity during REM sleep too. Related to this, it would be very interesting to see what other parts of the brain are doing during REM sleep, specifically how the activity may relate to the strong, irregular SN events in

claustrum – only if the authors happen to have simultaneous recordings from other areas during the recordings that they already made from the claustrum.

This is an important question, which we plan to examine in the future. Of particular interest are the projections to cortical areas. We have only anecdotal data during sleep at this point and they have not yet been analyzed in depth.

One other comment relates to the topic of competition between hemispheres. It's not clear to me that the alternating leading/lagging activity in the claustrum during REM sleep suggests a competitive process per se between the claustrum of each hemisphere. This is because the lagging claustrum's activity closely matches the leading one's with a short lag, and that does not necessarily imply competition, at least in the way that I usually think of neural competition (this is a minor point, but something that made we wonder what kind of competitive process was suggested while reading the first part of the paper before the lmc experiments). The clearest evidence of competition was the alternating activity between the two lmc's.

We apologize for a lack of clarity. The reviewer is right and this is what our results mean: the inter-claustrum competition, which was our initial observation, is indeed, as we realized later, only a reflection of competition in the midbrain; but the idea of potential competition between sides/claustra was, as the reviewer correctly notes, linked to the fact that the leading side (ie that which is active earlier) imposes its activity on the other. We agree that it was not necessarily the only potential explanation, but it seemed to us to be a likely one, and guided us in our search for its source. We have tried to clarify the text. For example (bottom of p4):

"This combination of matching bilateral pairs of waveforms, fixed positive or negative lags with the same absolute value, and temporal lead of the stronger side, suggested the possibility of competition between the two claustra, in which the stronger side at a given moment imposes its output on the weaker one, with a delay. By this hypothesis, leadership would depend on instantaneous variations of the relative "strengths" of activity on the two sides and the delay could be due to signal propagation and synaptic transfer in mirror-symmetric circuits. The suppression of the weaker side's own output—and its replacement by the stronger side's— suggested a winner-take-all type of bilateral competition."

Related to this, I think the scatterplot in Ext. Data Fig 6i might be clearer if only the periods of REM sleep were included, then perhaps a negative correlation would be more visible. It may also help to normalize the firing rates during each REM episode before compiling them across episodes.

We thank the reviewer for this suggestion. Indeed, analyzing only the periods of REM sleep yields a negative Pearson's coefficient of correlation. We have updated Ext. Data Fig 6i and h. We also explored normalizing the firing rates, but the results did not change substantially.

Overall, this is very nice work on a new model system for studying fundamental sleep-related processes during the main sleep states of REM and NREM sleep.

With many thanks.

Referee #3 (Remarks to the Author):

In this article Fenk et al. studied the neuronal networks and electrophysiological mechanisms of sleep in the bearded dragon lizard. They notably describe a new type of wave, the Sharp Negative (SN)

waves, occurring during a mammalian REM sleep like state recorded in the equivalent of the claustrum. The authors nicely demonstrate, that in contrast to the sharp waves of the NREM sleep like state, SN waves occur in each hemisphere with a stable 20ms delay between the left and the right hemisphere. Each hemisphere leading alternatively under the control of a GABAergic midbrain nucleus (the lmc) suggesting a competition between the two hemispheres to generate these waves.

This new neuronal mechanism clearly suggests that functional difference exist between the two sleep states of the lizard. This work is highly relevant in a comparative context to understand which mechanism and function are species specific and how sleep evolved. Indeed, by describing in detail this phenomenon, its associated neuronal network and this unexpected interhemispheric competition occurring during the possible REM sleep like state, Fenk et al. provide a new basis to draw parallels and reveal difference between species, which is crucial to understand the species specific and core functions of sleep. Many sleep-related brain oscillations have been implicated in processing memories in mammals, but little is known about the role brain oscillations play in this process in non-mammals. Consequently, this work on dragons opens a new opportunity to examine this question from a comparative perspective. In addition, this discovery would be of immediate interest to the field as the comparative approach, as there is a trend in the field to move away our mammalian centric view of sleep.

We are grateful to the reviewer for these comments and appreciation.

The authors use up-to-date methods for recording and analyzing the data. The figures are extremely clear and illustrate nicely all the different steps at different timescales from individual to average data. I also found that the experiment is well conceived demonstrating the existence of a new type of brain waves during sleep in lizard (SN waves), their generation and modulation. The interhemispheric competition described in the paper is highly stereotypical and convincing. The supplementary material showing individual data and its repeatability across individuals is very useful. The immunostaining and brain lesioning look clear and specific. In conclusion, I find the methods detailed enough to replicate the data.

Thank you.

My main concern about the paper is not related to the quality and reliability of the methods and results, but rather on the terminology and the broad interspecies interpretation. Indeed, whereas the mechanistic interpretation is clear and strong, the authors use mammal-based terminology (NREM and REM sleep) when describing the sleep states and electrophysiological features of sleep in dragons. Despite the fact that this group already published articles on sleep in lizards with such terminology, it is still not clear at what point the two states found in the bearded dragon are homologous / analogous or similar to the mammalian sleep states, as large differences remain between the two groups (See Blumberg et al 2020, Libourel & Barrillot 2020). This is particularly important for the specific terminology of the sleep waves, like sleep spindles, Sharp Wave ripples complex, delta waves or slow waves. All these waves refer to specific neuronal networks, associated with specific function, like memory consolidation, and none of them strictly fit with the waves found in reptiles in term of shape, localization and frequency. Using mammalian terminology in non-mammalian species based on similar morphology alone is clearly not enough to infer the homology the use of the same term implies. and the use of mammalian terms clearly biases the cross-species interpretation of the data, and undercuts the true power of comparative research.

To avoid the problems associated with using mammal-based names in non-mammalian species, the authors should find a way to revise the terminology (NREM, REM SWr). The author could refer to "REM sleep" as "REM sleep like state" or "NREM sleep" to "reptilian NREM sleep", and "SWr" as "SWr-like" or "reptilian SWr", in the whole text, abstract and title. I actually noticed that the author did

this for the waves found during the REM sleep like state by naming them SN. As an example, if these waves would have been named Slow oscillations (SO) because they look like mammalian SO, this would have completely changed our interpretation of the lizard REM sleep like state and possibly our interpretation of the evolution of sleep. By no means does this criticism of the terminology undercut the importance of the data presented.

We thank the reviewer for raising these points. We are, like him, conscious of the historical baggage that terms carry with them in comparative neuroscience, and notably with sleep, since most if not all of the terminology originated with work in mammals decades ago. Anything that one identifies now in a non-mammal must therefore take this history into account and somewhat accommodate itself around it.

The problem that one is confronted with is to decide whether existing/mammalian terminology is, by definition, only applicable to mammalian systems, or whether a broader use might be allowed, and if so, for what degree of demonstrated equivalence. As the reviewer knows, it is a real and non-trivial problem for phenomena such as those which we study. Some people we have encountered, for example, reject the use of the term “sleep” to characterize non-mammalian sleep. Some reject the use of the term “claustrum” in a reptile, despite its demonstrated transcriptomic homology (Norimoto, Fenk et al., 2020), because reptilian “claustrum” does not connect to a neocortex (expectedly since reptiles do not possess a neocortex). The terminology debate (not specific to sleep, neuroscience or evolutionary biology) thus seems to rest on differing levels of acceptance of differences.

Our approach in naming waveforms and field potentials is practical, descriptive or “morphological”, and based on the features of the claustral and cortical recordings which we detailed in earlier publications. We infer or presuppose neither homology nor identity of function. Those are critical questions, but we remain agnostic: they remain open and they will hopefully be answered as we learn more about those different systems. This is, we believe, our common objective.

Because our approach to nomenclature is “morphological”, we could not apply usual (i.e., mammalian) terminology to what we here call SNs: for example, SNs do not match mammalian SOs, being neither slow nor periodic, and they do not occur in the sleep phase in which one might have expected them to. For these reasons, we estimated that these events required a new name. We chose SNs but would be happy to use better alternatives. What we name “SWRs”, by contrast, have many of the morphological characteristics of SWRs described in mammalian hippocampus (but also in other regions), and are present in slow wave sleep; this is why we use this terminology. As we indicate above, we do not infer any homologous or functional implications to this terminology, though we understand that readers versed only in mammalian hippocampal literature might make such inferences. We try to be as clear as possible and have tried to clarify this further in our manuscript.

*Our approach in naming sleep states is of course inspired by mammalian terminology and to the extent that *Pogona* sleep has two clear alternating sleep states, one characterized by slow waves and what we identify as SWRs, the other with faster forebrain activity similar to that recorded during awake states and rapid eye movements (among other features), we have since our initial findings (Shein-Idelson et al., 2016) called them slow-wave and REM sleep respectively.*

I have also several other comments on the papers listed below:

- From line 49 to 60 the introduction on sleep in reptiles should be fully revised accordingly to my main comment regarding terminology.

In response to the reviewer’s request, we have clarified our definitions in the introduction as suggested. Those lines now read:

“In the Australian bearded dragon *Pogona vitticeps*, a diurnal agamid lizard, the two phases of a regular electrophysiological sleep rhythm (recorded in the dorsal-ventricular ridge or DVR) consist of alternating local-field-potentials dominated by sharp wave ripples occurring irregularly every 0.5-2s, and faster awake-like activity that co-occurs with rapid-eye movements¹⁹. Because of the similarities between these two electrophysiological sleep phases and mammalian SW and REM sleep respectively, we identify them as non-REM-like and REM-like activities (thereafter named NREM and REM, respectively). (Note that our nomenclature is descriptive, and does not necessarily imply, for lack of knowledge at this point, functional or mechanistic similarity with mammalian sleep states.) These two activity modes alternate regularly throughout the night with a period of 1.5 to 2.5 minutes¹⁹.”

- Line 67: the authors specify that the recordings were made from the claustrum equivalent or the adjacent anterior DVR. Are there any differences between the recording sites in terms of SN, spikes, delay?

We could detect SNs in all of our claustrum recordings, with a trend towards larger SN amplitude at its lateral and fiber-dense border with the anterior DVR. L-R delays for all recordings were within the same, narrow range (cf. Fig. 2d). There are no discernable delays along a probe shank, as illustrated below and in response to the reviewer’s next question. We did include a single recording where we targeted the claustrum bilaterally, but where the tip of one of the two probes ended up just outside the claustrum, in the bordering region with the anterior DVR. These data are also included in Fig. 2d, and followed the general pattern.

Also note that previous experiments suggest that DVR activity becomes increasingly different the farther away from the claustrum one records. We plot below a short period of REM, taken from the data shown in Fig. 1 of Norimoto, Fenk et al. (histology in Extended Data Fig. 1h). Clear SN waveforms are detectable in the claustrum, but not in the LFP recorded at a distance in the ipsilateral anterior DVR, although some degree of correlation exists between the two recordings. These differences and the potential propagation of claustrum REM activity to downstream areas such as the DVR are interesting questions warranting future work.

- How much does the LFP signal vary as a function of depth? A representation of the LFP traces from the different electrode sites and brain regions would be very useful. This could illustrate if there is any phase reversal recorded from the silicon probe showing a local phenomenon like that shown for the reptilian SWr (Shein Idelson et al 2016). The author should provide more information about the regional aspect of the SN.

We attach below a figure that shows a representative example of a single SN waveform, recorded in the two claustra (a), and taken from the short epoch depicted in Fig. 2c. Plotted is the LFP (<100Hz)

recorded at 16 electrode sites of a 32-channel silicon probe (skipping every second site for illustration). The deepest channels are at the bottom; they were located in the claustrum, and are the ones we used for further analysis. Note the gradual decrease in SN amplitude towards more superficial channels, the synchrony across channels, and the apparent absence of a waveform reversal. The most superficial channels (in faint gray) may be outside the claustrum, possibly even within the ventricle.

Note also that, even though we have shown previously that SWRs are generated “locally” within the claustrum (Norimoto, Fenk et al. 2020), we have not observed phase reversals in our *in vivo* claustrum recordings (consistent with the absence of layered architecture akin to that of areas CA1 or CA3 in the mammalian hippocampus).

Shown below in (b) is a corresponding *lmc* waveform from the data plotted in Fig. 4e,f. Plotted is the LFP (<100Hz) recorded at 21 electrode sites from a Neuropixels probe inserted into the *lmc*. The deepest channel is shown at the bottom, and was located in the caudal end of the *lmc* (cf. Fig. 4b). Note again the gradual decrease in amplitude towards more superficial channels, and the absence of a discernible delay or phase reversal.

- It would be very informative to provide more information regarding the SN extracellular unit discharge. Over all the animals and electrode sites, how many SN were phase locked with unit activity. It is actually not clear me to how reliably the neuronal discharge is linked with the SN. Is the figure 1 panel e showing only one electrode site from one animal? This should be clarified.

We thank the reviewer for this request. Fig 1e shows the distributions of spike times for 4 single units, relative to the time of peak $[dV/dt]$ of the SNs. The spikes are from one animal and one night, recorded

during 7 hours of sleep ($n = 100,632$ events). The probability of these units producing at least 1 spike: 14-43%; probability of these units producing more than 1 spike: 0.3-3%. Note, these numbers depend heavily on our ability to sort single units. This remains a challenge, especially during periods when SNs happen at higher frequency, and when nearby units fire spikes within a few milliseconds. Spikes were recorded using a 32-channel silicon probe (NeuroNexus; 50- μm pitch, 177- μm^2 surface area for each site; in 2 rows of 16 contacts). The spikes from each unit are recorded on multiple sites along the linear shank. We show below, for illustration, the average spike waveforms of two of those units (green and black, left), recorded on 5 channels each (waveforms exported from the ironclust spike-sorting software). We also plotted, as illustrative examples, two short episodes of REM sleep, from two animals with aligned claustrum LFP (red) and claustrum unit spikes.

Inserted below are the distributions of spike times for 10 units from this animal 2 dataset, relative to the time of peak $[dV/dt]$ of the ipsilateral SNs (left column), as in Fig. 1e ($n = 128,453$ events). Note the reliable increase in firing, locked to the descending phase of the SNs.

We also added the spike times relative to SNs ($n = 123,798$) detected in the contralateral claustrum (right column). Note the peaks around -20ms and +20ms, corresponding to periods when the contralateral and ipsilateral side are leading, respectively. Also note that peaks are higher when the contralateral claustrum is leading (the side where the spikes have been recorded), consistent with greater synaptic drive and larger amplitude SNs in the dominant hemisphere (cf Fig. 2c-f).

We added this panel to our Extended Data Fig. 1.

Extended Data Fig. 1e. Extracellular SN potentials coincide with phasic and synchronized firing in claustrum units. Data as in Fig. 1e, but data taken from a different animal, with bilateral claustrum recording. Top: 128,453 (left) and 123,798 (right) SNs together with the mean waveforms (bold). Histograms: distributions of spike times in 10 units recorded in L claustrum, relative to the time of peak $|dV/dt|$ of the SNs (red stippled line) detected either ipsilaterally (left) or contralaterally (right). Right column: Note peaks around -20ms and +20ms, corresponding to periods when the contralateral and ipsilateral side are leading, respectively. Also note that peaks are higher when the contralateral claustrum is leading (the side from which the spikes shown are recorded), consistent with greater synaptic drive and larger amplitude SNs in the dominant hemisphere (cf Fig. 2c-f).

- One of the major findings of this paper is the inter-hemispherical coordination of the SN during REM sleep like state, in opposition to the SWr of NREM sleep like state which are not synchronize across hemisphere. This is an important feature to assume that the SN and SWr are generated by different neuronal processes, possibly underlying different function. However, only figure 2 panel b illustrated that purpose. But how many animals, nights, samples do this correspond to? We have actually no idea about the reliability of the absence of coordination. It would help to provide more information on this.

We thank the reviewer for this question; we agree that this difference between activities during REM and NREM sleep is an important result of our study. The fact that SWRs are uncoordinated between hemispheres is consistent with our previous work (Norimoto, Fenk et al., 2020), showing that each claustrum can generate SWRs independently of the other (after unilateral claustrum lesions, *in vivo*), and even in complete isolation (*in ex vivo* slice preparations). However, this result did not preclude the possibility of observing inter-hemispheric coordination during sleep in intact animals, which we address in the current study.

Fig. 2b shows a short stretch of NREM sleep (top), together with superimposed LFP sweeps recorded from one side (and their average), triggered on 100 SWRs detected on the other side (bottom). These data are from the recording from which Fig. 2a is also drawn.

To show that the absence of coordination between SWRs in L and R claustra is a reliable phenomenon, we analyzed the data from 5 additional animals and 9 hrs of sleep each, including data from 2 animals with a unilateral *lmc* lesion. We have now added the results to (new) Extended Data Fig. 2 (and also below), confirming the absence of coordination between SWRs in L and R claustra under our experimental conditions (natural sleep in adult animals, at room temperature).

Extended Data Fig. 2b and c. SWR production in the claustrum is not coordinated on L and R sides during NREM sleep. **b.** Averages (blue) of 1000 SWRs randomly picked through 9hrs of sleep, together with the SWR-triggered average LFP (red) on the contralateral side (shaded areas: standard deviation). **c.** Corresponding auto- and cross-correlograms for animals I-V in b. Note flat cross-correlograms.

- The *lmc* lesion experiment is highly convincing. I wonder if this lesion also influences the reptilian SWr occurrence, amplitude?

It is difficult to answer this interesting question at this time: SWR frequency and amplitude vary across animals and between recording sessions within individual animals. To answer the reviewer's question conclusively, recordings from CLA would have to be made both before and after *lmc* lesions, which is extremely difficult.

As the authors state in the discussion, I also wonder whether this *lmc* Claustra-coordination exists during wake. As the authors recorded for hours even after lights on, I wonder whether they have

enough wake to provide this information? This would be indeed of great interest to see if differences exist between wake and the REM sleep like state.

This is a very good point and one which we are following up. The answer is complex since wake states are far more diverse than sleep states. Our data at present are too preliminary to make any clear statement.

- The author should define WTA and STDP, as the abbreviations are used without definition.

We have corrected this.

- At what temperature were the animals recorded?

At constant room temperature of ~ 21.5°C. This is now indicated in the methods section.

- In the SN detection and matching methods, how was noise estimated (line 423).

*We provided a short description of the noise estimation (line 425 in original ms). In the interest of clarity, we have rewritten this definition to read:
"To estimate the distribution of noise, i.e. of small LFP deflections wrongly identified by our method as SNs, we multiplied the signal by -1 and repeated the same process of triplet peak detection. This was equivalent to attempting to detect sharp positive deflections, which did not exist in the signal. Consequently, any positive events detected by our triplet peak detection were the result of LFP noise. We then used those wrongly identified positive events to establish minimum thresholds of amplitude and duration on the originally detected SNs."*

- Line 403: how were the channels of the probes chosen?

We typically chose the deepest channel, whose anatomical location was identified by electrolytic lesion and/or selective application of Dil on the electrode shank.

- Fig 4c (line 741) where are the "corresponding levels"? This is not clear.

Apologies for the lack of clarity. Corresponding to those in b. This has been clarified in the legend.

- The following references are missing but highly relevant to the paper:

1- Wang et al 2020, "Subdivisions of the mesencephalon and isthmus in the lizard Gekko gekko as revealed by ChAT immunohistochemistry",

2- Ghosh et al 2022, "Running speed and REM sleep control two distinct modes of rapid interhemispheric communication"

In particular Ghosh et al 2022 should be discussed regarding the finding of the paper.

Wang et al 2020 seems to contain some errors. For example: their summary in Fig. 6f, labels the lsm (=lmc) as a cholinergic nucleus, which it is not: it is GABAergic. In Table 1, Wang et al. suggest that turtle/lizard lmc/lsm is equivalent to avian lpc and mammalian PBN, again in contradiction with our results.

Ghosh et al 2022: Thank you for pointing out this study: we now mention it in the discussion:
“Antiphase activity between the two hemispheres has recently been identified in running rodents and in REM with fast rhythms (140Hz “splines”) in superficial retrosplenial cortex(Ghosh 22, #60), but the mechanisms underlying this relationship are unknown.”

References

- 1 Ramón y Cajal, S. *Histologie du système nerveux de l'homme & des vertébrés*. Vol. 2 (1911).
- 2 Wang, Y., Major, D. E. & Karten, H. J. Morphology and connections of nucleus isthmi pars magnocellularis in chicks (*Gallus gallus*). *J Comp Neurol* **469**, 275-297, doi:10.1002/cne.11007 (2004).
- 3 Marin, G. et al. A cholinergic gating mechanism controlled by competitive interactions in the optic tectum of the pigeon. *J Neurosci* **27**, 8112-8121, doi:10.1523/JNEUROSCI.1420-07.2007 (2007).
- 4 Goddard, C. A., Sridharan, D., Huguenard, J. R. & Knudsen, E. I. Gamma oscillations are generated locally in an attention-related midbrain network. *Neuron* **73**, 567-580, doi:10.1016/j.neuron.2011.11.028 (2012).
- 5 Mysore, S. P. & Knudsen, E. I. A shared inhibitory circuit for both exogenous and endogenous control of stimulus selection. *Nat Neurosci* **16**, 473-478, doi:10.1038/nn.3352 (2013).
- 6 Sorenson, E. M., Parkinson, D., Dahl, J. L. & Chiappinelli, V. A. Immunohistochemical localization of choline acetyltransferase in the chicken mesencephalon. *J Comp Neurol* **281**, 641-657, doi:10.1002/cne.902810412 (1989).
- 7 Gonzalez-Cabrera, C., Garrido-Charad, F., Roth, A. & Marin, G. J. The isthmic nuclei providing parallel feedback connections to the avian tectum have different neurochemical identities: Expression of glutamatergic and cholinergic markers in the chick (*Gallus gallus*). *J Comp Neurol* **523**, 1341-1358, doi:10.1002/cne.23739 (2015).

Reviewer Reports on the First Revision:

Referees' comments:

Referee #1 (Remarks to the Author):

Thank you for your responses. Indeed, this is a nice neurophysiology paper.

Referee #2 (Remarks to the Author):

The authors have addressed my comments. This will be a beautiful addition to our knowledge of the neural activity and mechanisms of sleep, and will contribute to our understanding of sleep in both this species and across species.

Referee #3 (Remarks to the Author):

The authors answered properly to all of my comments.

however, I am still convinced that using the same mammalian vocabulary (like NREM, REM sleep, and SWr, in particular) undermines the whole purpose of comparative research. Moreover, the authors agreed with the fact that using such names does not indicate homology or functional similarity, and added a sentence to that affect in the introduction. So why continuing using them if it could create confusion in the interpretation?

I still think that the authors should not use REM, NREM and SWr, as so many differences still remain with mammalian sleep. Comparative biology is the only way to bring a wider view on the core role of sleep. It is a difficult task, particularly because two morphologically similar characters could be generated by different neuronal networks and could sustain other functions in another species. I agree with the authors that the words used have a first morphological and descriptive sense. However, in the minds of non-specialists and even sleep specialist, they also relate to specific mammalian functions (like memory consolidation for SWr) or states (like dream association for REM sleep), which are clearly far from been demonstrated in lizards. Using REM-L (for REM like), NREM-L and SWr-L, in the WHOLE text, including the title, is the good compromise to keep the message without denaturing the richness of the discovery.

PA Libourel